# Early lineage segregation of primary myotubes from secondary myotubes and adult muscle stem cells

Gauthier Toulouse [1], William Jarassier[2,4], Sabrina Jagot [1,4], Valérie Morin[1], Fabien Le Grand[1,2] ✉ & Christophe Marcelle [1,3] ✉

Myogenesis in amniotes occurs in two waves. Primary myotubes express slow myosin (often with fast myosin) and likely act as scaffolds for secondary myotubes, which express only fast myosin. The embryonic origins and relationships of these lineages, and their connection to satellite cells, remain unknown. Here, we combine a TCF-LEF/β-catenin signaling reporter with precise in vivo electroporation in avian embryos to trace limb muscle progenitors from early migration to fetal stages. We identify two distinct progenitor populations that coexist from the onset: reporter-positive cells give rise exclusively to primary myotubes, while reporter-negative cells generate secondary myotubes and satellite cells. We also reveal a previously unrecognized role for TCF-LEF/β-catenin signaling in spatially organizing the primary lineage via Cxcr4-mediated control of myoblast migration. These findings redefine the developmental origin of myogenic lineages, resolve a long-standing question in muscle biology, and provide a molecular framework for investigating how muscle fiber diversity emerges and how distinct lineages contribute to the functional specialization of skeletal muscle.

The early steps of skeletal muscle morphogenesis have been extensively studied, leading to the emergence of detailed models of the cellular and molecular mechanisms regulating the first steps of myogenesis[1–8]. One of the lesser-known steps of myogenesis is the generation of so-called "primary" and "secondary" myotubes. Microscopists studying muscle formation in vertebrate (birds and rodents) embryos half a century ago have identified two distinct waves of myogenesis. The first, taking place during the embryonic phase of development, generates what they have named primary myotubes. The second wave, observed in fetal life, results in the emergence of secondary myotubes. Distinct morphological characteristics separate primary and secondary myotubes[9,10]. Primary myotubes are initially large and extend from tendon to tendon of embryonic muscles. Secondary myotubes are initially small and tightly organized around primary myotube[9,11,12]. Such an association led to the widely held but

unproven assumption that primary myotubes may serve as an organizing scaffold for the morphogenesis of the secondary lineage[10]. Primary myotubes form before motoneurons reach the muscle masses, and they develop normally until birth despite denervation or inhibition of neural activity[13,14]. Secondary myotubes, on the contrary, form after motoneurons have reached the muscle masses and most authors agree that the emergence of secondary myotubes is dependent on innervation[13–18]. Biochemical properties also distinguish primary and secondary myotubes. In rodents and birds, primary myotubes are characterized by their expression of slow myosin (in variable combination with embryonic/fast myosin), whereas secondary myotubes express exclusively embryonic fast myosin[19–22]. Crucially, this initial, genetically encoded, fiber type pattern is plastic and modulated throughout life by factors such as innervation, exercise, and hormonal influences[10].

[1]Université Claude Bernard Lyon1, MeLiS Laboratory, CNRS, INSERM, Lyon, France. [2]Université Claude Bernard Lyon1, PGNM Laboratory, CNRS, INSERM, Lyon, France. [3]Monash University, Australian Regenerative Medicine Institute, Clayton, VIC, Australia. [4]These authors contributed equally: William Jarassier, Sabrina Jagot. ✉e-mail: fabien.le-grand@univ-lyon1.fr; christophe.marcelle@univ-lyon1.fr

Studies on the cellular origin of primary and secondary myotubes have identified three successive waves of myoblast populations: embryonic, fetal and adult (or satellite cell) myoblasts[10]. Embryonic myoblasts are isolated from muscle masses during the first week of development in chicken and up to the second week in mouse; fetal myoblasts thereafter[23]. The adult/satellite cell population settles under the myofibers' basal lamina shortly before hatching or birth and serves as a reserve stem cell population for muscle growth and repair after birth and into adulthood. In vitro, embryonic and fetal myoblasts exhibit intrinsic differences in fusion ability, proliferation and response to growth factors[10,24–28]. Furthermore, transcriptomics analyses of these two populations confirmed that they possess distinct genetic programs[29,30].

Nonetheless, the lineage connection between these two populations of myoblasts remains a subject of unresolved matter[10,31]. Limb grafting in chicken embryos suggested that two successive waves of myogenic progenitors, each with distinct characteristics, migrate into the limb bud to form primary and secondary myotubes[32]. In vitro studies have further complicated the issue, yielding conflicting hypotheses that propose either a single evolving population, transitioning from displaying embryonic to fetal characteristics[28,29,33] or two distinct populations present in the limb from the onset of its formation[10,25,34].

Here, we have analyzed the developmental path that muscle progenitors of chicken embryos follow from the time of their migration away from the wing somites into the limb bud mesenchyme until late fetal life, shortly before hatching. Using the in vivo electroporation technique to specifically target the muscle progenitor population and a transcriptional reporter to monitor TCF-LEF/β-catenin dependent signaling activity in real time, we have discovered that TCF-LEF/β-catenin activation distinguishes positive and negative muscle progenitors within the early growing wing bud. Biochemical analyses showed that the reporter's activity is restricted to early stage (PAX3$^+$/PAX7$^+$/MYF5$^+$/MYOD$^-$) of myoblast differentiation and confined to the early phases of embryonic development, while absent thereafter until hatching. We designed a novel, dynamic Tet-on-based tool (named TCF-Trace) to follow the fate of reporter-positive myogenic progenitors. This demonstrated that all reporter-positive myoblasts readily differentiated into primary myotubes, while the reporter-negative myoblasts gave rise to the late-forming, secondary myofibers, and also to satellite cells. Next, we demonstrated that TCF-LEF/β-catenin signaling does not regulate cell proliferation and differentiation but rather plays a crucial role in the spatial distribution of limb primary muscle progenitors. Transcriptomic profiling of the two progenitor types in chicken confirmed the existence of two distinct co-existing lineages. Remarkably, strong similarities between species suggest a conserved developmental program, with Wnt signaling tightly associated with primary myogenesis in both chicken and human. Finally, we show that Wnt signaling likely influences progenitor distribution via transcriptional control of the migration regulator CXCR4.

These discoveries not only address a long-standing question in muscle biology but also provide a crucial molecular gateway for understanding how these early developmental processes contribute to the fiber type diversity of muscle fibers.

## Results

### TCF-LEF/β-catenin dependent signaling is restricted to early limb muscle development

We recently generated avian transgenic lines that revealed unexpected features of TCF/β-catenin-responding cells and tissues[35]. Among those tissues, we were intrigued by the presence of cells activating the pathway within the limb muscles masses. We decided to investigate further this observation. All limb muscles originate from a single origin, the lateral part of the somite (named the ventro-lateral lip - VLL)[1],

which can be easily targeted using the electroporation technique[36]. This contrasts with trunk muscles that originate from various regions of somites[5,6]. The VLL therefore constitutes an attractive experimental model to study all steps of muscle formation and patterning (Supplementary Fig. 1). To verify that TCF/β-catenin-dependent signaling was active in the myogenic lineage, we have electroporated the transcriptional reporter of TCF/β-catenin-dependent signaling used previously[35] in the VLL of somites 16–21 of E2.5 chicken embryos[1,4], from which limb muscle progenitors originate (Fig. 1a; Supplementary Fig. 1). The transcriptional reporter, named 16TF-VNP, comprises sixteen TCF/LEF binding sites upstream of a destabilized, nuclear Venus fluorescent protein, surrounded by translational enhancers to boost its activity (see "Material and Methods"; Fig. 1b). This and all electroporated constructs used in this study were cloned into expression vectors containing two Tol2 sequences (T2: transposable elements from medaka fish[37]) surrounding the entire construct to allow their stable integration in the chicken genome in the presence of transposase, provided by a co-electroporated transposase plasmid (CAGGS-Transposase). Since electroporation leads to the mosaic transfection of the targeted cell population, the reporter was co-electroporated with a ubiquitously-expressed fluorescent marker to identify and analyze electroporated cells individually. We followed the activity of the TCF/β-catenin-responding cells in the myogenic lineage throughout embryonic and fetal development.

At E3, i.e. twelve hours after electroporation, the migration of progenitors emanating from the VLL has started (Fig. 1c, d). At that stage, two distinct locations of TCF/β-catenin-responding cells were observed: a strong expression in a majority (66%) of electroporated epithelial cells located within the VLL and a weaker expression in a minority (21%) of migrating, electroporated cells exiting from the VLL (Fig. 1c, k; Supplementary Fig. 2a, b). The expression of PAX7 in all (16TF-VNP$^+$ and 16TF-VNP$^-$) electroporated migrating cells confirmed that these are bona fide muscle progenitors (Supplementary Fig. 2a–c). Half a day later, at E3.5, as all muscle progenitors have exited the VLL[38], 45% of electroporated muscle progenitors were strongly positive for the 16TF-VNP reporter (Fig. 1e, k). During the following three days of development (E4.5 to E6.5) this proportion and level of expression remained relatively stable, in about 50–55% of electroporated cells (Fig. 1f–h, k). During this time period, 16TF-VNP$^+$ cells were distributed among 16TF-VNP$^-$ cells, with an increasing tendency towards a localization at the distal end of the muscle masses as development proceeded (Fig. 1f–h). 16TF-VNP$^+$ cells were evenly distributed along the dorso-ventral axis of muscles and were also present in the ventral muscle mass of the limb bud (Supplementary Fig. 2d–f). A sharp decrease in the reporter's activity was however observed on the next day, at E7.5, as 16TF-VNP$^+$ cells became sparse and were confined mainly to the distal-most part of the muscle masses (Fig. 1I, arrowheads).

We then performed long-term analyses of the reporter's activity. Embryos electroporated at E2.5 were analyzed on transversal and longitudinal sections of limbs collected at E9.5, E12.5, E14.5, E16.5 and E18.5. Despite a massive labeling of the muscle masses by the ubiquitously expressed electroporation marker, from E9.5 - E18.5, we did not detect any 16TF-VNP$^+$ cells at any of the analyzed developmental stages (Fig. 1j; Supplementary Fig. 3 and Supplementary Fig. 4).

These data demonstrate that TCF/LEF transcription in the muscle progenitor population is dynamic (see Fig. 1l). An important feature of the reporter's activity is that it was observed in about 50% of electroporated muscle progenitors from E3.5 until E6.5. During that time period, muscle progenitors differentiated into many multinucleated muscle fibers, the first visible at around E5/E5.5[36]. As muscle differentiation progressed and distinct muscle bundles emerged, the reporter activity sharply dropped and was kept silent until hatching.

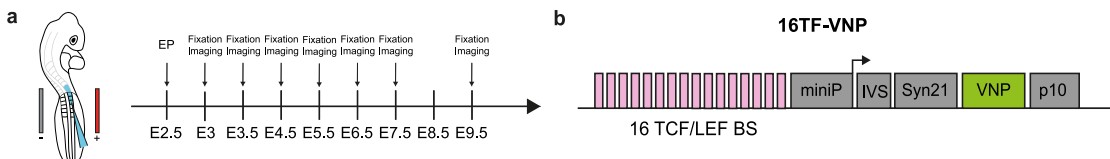

**Nature Communications**| (2025)16:7858

### TCF-LEF/β-catenin positive cells are PAX3⁺/PAX7⁺/MYF5⁺/MYOD⁻ muscle progenitors

We then investigated the myogenic differentiation state of TCF-LEF/β-catenin activating cells. During myogenesis, muscle progenitors sequentially express different transcription factors that correspond to different phases of myogenic commitment[8,39,40]. In the mouse and chicken embryos, the proliferative muscle progenitor population comprises PAX7⁺MYF5⁻ slow-dividing and PAX7⁺/MYF5⁺ fast-dividing cells[41]. MYOD expression signals the exit of progenitors from cell cycle, and MYOG expression corresponds to terminally differentiated, pre-fusing muscle cells[42].

In birds, PAX7 and PAX3 proteins are co-expressed in limb myogenic progenitors, from the moment they exit the VLL and migrate into the limb mesenchyme[43,44]. In fact, PAX3 and PAX7 co-expression

**Fig. 1 | TCF-LEF/β-catenin dependent signaling is restricted to early limb muscle development. a** Brachial somites were electroporated at E2.5 and embryos were analyzed at indicated timepoints. **b** Representation of the transcriptional reporter (16TF-VNP) used to monitor TCF/LEF/β-catenin dependent signaling. 16 TCF/LEF binding sites (BS) were placed upstream of a minimal promoter (miniP) driving the expression of a nuclear, destabilized Venus fluorescent protein (VNP); three translational enhancers were added (IVS, Syn21 and p10) to boost protein production. **c–e** Dorsal view of confocal stacks of brachial somites electroporated with a ubiquitously expressed dTomato and the 16TF-VNP, observed at E3 and E3.5. E3 VLL timepoint represents the epithelial electroporated cells still located in the VLL at E3 while E3 Migrat. represents the mesenchymal, migrating electroporated cells at E3. Somite borders are indicated by dotted lines; **d** is an enlargement of (**c**).

**f–i** Dorsal views of confocal stacks of limb buds observed between E4.5 and E7.5, electroporated with either a ubiquitous TagBFP (**f–h**) or a ubiquitous nuclear dTomato (**i**), together with the 16TF-VNP. Arrowheads in (**i**) indicate the last remaining 16TF-VNP⁺ cells present at E7.5. **j** Transversal section of E9.5 limb bud electroporated with a ubiquitous nuclear dTomato together with the 16TF-VNP. **k** Quantification of the percentage of 16TF-VNP⁺ cells between E3 and E6.5, each dot represents a limb bud (*n* = 5, 5, 4, 9, 5 and 6 for each time point, respectively). **l** Schematic representation of 16TF-VNP activity in the myogenic lineage during development. Box plots show the median (center line), 25th and 75th percentiles (box limits), and whiskers extending to the smallest and largest values within 1.5 x IQR. Scale bars: 100 µm (**c–f**) or 200 µm (**g–j**).

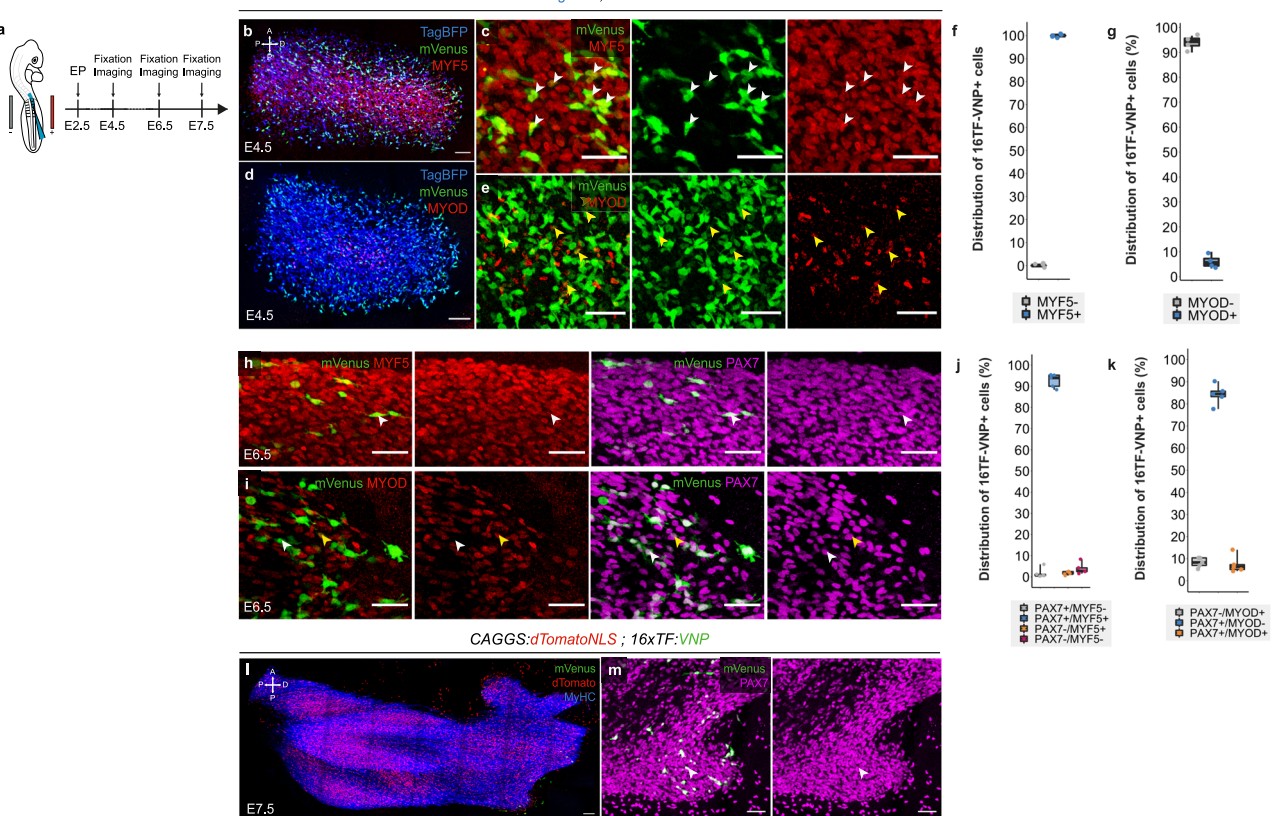

*CAGGS:TagBFP ; 16xTF:VNP*

*CAGGS:dTomatoNLS ; 16xTF:VNP*

**Fig. 2 | 16TF-VNP⁺ cells are early myogenic progenitors. a** Brachial somites were electroporated at E2.5 and embryos were analyzed at indicated timepoints. **b–e** Dorsal views of confocal stacks of E4.5 limb buds electroporated at E2.5 with an ubiquitous TagBFP, the 16TF-VNP reporter and immunostained for MYF5 (**b, c**) or MYOD (**d, e**). White arrowheads in (**c**) indicates 16TF-VNP⁺/MYF5⁺ cells; yellow arrowheads in (**e**) indicate some of the few 16TF-VNP⁺/MYOD⁺ observed. **f, g** Quantification of the percentage of 16TF-VNP⁺ cells positive for MYF5 (**e**) and MYOD (**f**) each dot represents a limb bud (*n* = 5 and 4, respectively). **h, i** Dorsal view of confocal stacks of E6.5 limb buds electroporated with a ubiquitous TagBFP, the 16TF-VNP reporter, and stained for PAX7 and MYF5 (**h**) or PAX7 and MYOD (**i**). The TagBFP channel is not represented. The white arrowheads indicate 16TF-VNP⁺/

PAX7⁺/MYF5⁺ cells (**g**) and 16TF-VNP⁺/PAX7⁺/MYOD⁻ ones (**i**). The yellow arrowhead in (**i**) indicates a 16TF-VNP⁺/PAX7⁺/MYOD⁺ cell. **j, k** Quantification of the percentage of 16TF-VNP⁺ cells positive for PAX7 and MYF5 (**j**) or PAX7 and MYOD (**k**), each dot represents a limb bud (*n* = 5 for both staining). **l, m** Dorsal view of confocal stacks of E7.5 limb buds electroporated with a ubiquitous nuclear dTomato, the 16TF-VNP reporter, and stained for PAX7. The arrowhead indicates one of the few remaining 16TF-VNP⁺ cells that also expresses PAX7 (*n* = 5 embryos). **m** is an enlargement of (**l**). Box plots show the median (center line), 25th and 75th percentiles (box limits), and whiskers extending to the smallest and largest values within 1.5 x IQR. Scale bars: 50 µm (**h, i, m**), 100 µm (**b–e**), or 200 µm (**l**).

persists in all limb myogenic progenitors throughout development, from E10.5-E16.5, i.e., when progenitors assume satellite cell positions along the myofibers under the basal lamina (Supplementary Fig. 5d–g)[6]. Therefore, all 16TF-VNP⁺ and 16TF-VNP⁻ progenitors present in the limb at E4.5 co-expressed PAX3 and PAX7 (Supplementary Fig. 5a–c).

To further characterize the molecular and proliferative signature of TCF-LEF/β-catenin activating cells in the limb bud, we performed immunostainings against PAX7, MYF5, MYOD and EdU. At E4.5, all muscle cells present in avian limb muscle masses are mononucleated

and express PAX7[36]. At this stage, we observed that while MYF5 expression was widespread throughout the entire muscle progenitor population, MYOD expression was restricted to its central region (Fig. 2b, e). We observed that all 16TF-VNP⁺ cells (100%) expressed MYF5, while only 6% expressed MYOD (Fig. 2b–g). At E6.5, many polynucleated MyHC⁺ muscle fibers are present throughout the growing wing and they are tightly intermingled with single-cell progenitors[36]. Similar to E4.5, we observed at E6.5 that the vast majority of 16TF-VNP⁺ cells (93%) co-expressed PAX7 and MYF5, but that most (85%) 16TF-VNP⁺ did not express MYOD (Fig. 2h–k). At E7.5,

even though very few progenitors expressed the reporter, all of them were PAX7[+] (Fig. 2l). Even though we have previously shown that all PAX7[+]/MYF5[+] progenitors are faster-dividing cells than PAX7[+]/MYF5[-] progenitors[41], it was possible that the 16TF-VNP[+] and 16TF-VNP[-]subpopulations proliferated at different rates. However, labeling of embryos with EdU showed that both the negative and the positive populations displayed the same rate of proliferation (Supplementary Fig. 6a–c).

Together, these experiments demonstrate that TCF-LEF/β-catenin dependent signaling is strictly restricted to a narrow time window of myogenesis, to a population of proliferating PAX3[+]/PAX7[+]/MYF5[+] progenitors. As soon as these progenitors progress further along the myogenic path, initiating MYOD expression and exiting cell cycle, TCF-LEF/β-catenin dependent signaling is turned off.

## TCF-Trace, a dynamic lineage tracing system to follow the fate of TCF-LEF/β-catenin[+] myogenic precursors

The 16TF-VNP reporter we engineered provides a snapshot of TCF-LEF/β-catenin dependent signaling at the time of analysis. It was, however, possible that limb muscle progenitors fluctuate between a 16TF-VNP[+] and a 16TF-VNP[-] state. To test this, we designed a reporter construct where the destabilized nuclear mVenus fluorescent protein was placed in tandem with a stable nuclear mCherry (half-life: about 18 h[45]). This technique has been used in *Drosophilia*[46] to test whether the activity of a promoter is fluctuating between an ON/OFF state. If the activity of the reporter was fluctuating, there would be more cells labeled with the stable mCherry than the destabilized mVenus. On the contrary, if cells constantly respond to the signal, only double-positive cells should be observed. The construct was electroporated in the VLL at E2.5 and examined in the limb at E4.5. We observed a near-perfect (98%) correlation of mVenus and mCherry stainings, indicating that, within the time frame of the experiment, the 16TF-VNP[+] progenitors maintain the reporter's activity and do not fluctuate between a positive and negative state (Supplementary Fig. 7a-c).

It therefore became important to investigate the long-term fate of myogenic progenitors that activate TCF-LEF/β-catenin dependent signaling. To address this, we developed a lineage tracing system using both Tet-On and Cre-Lox technologies (Fig. 3a). This system aims to permanently label cells with dTomato fluorescence when they simultaneously experience TCF-LEF/β-catenin signaling and are exposed to doxycycline (Fig. 3b). The Tet-on technology[47,48] is a well-established system of drug-induced gene activation, which displays low background and high induction rates, in particular when using the "rtTA/TRE 3 G" system[49,50]. Combining this to the CRE-mediated excision of "Stop/PolyA" sequences placed upstream of a fluorescent protein should deliver a very sensitive system to permanently label myogenic progenitors. However, a significant drawback of the Tet-on system is that the rtTA protein is very stable[51], making it unsuitable for dynamic studies. We hypothesized that the rtTA protein could be destabilized, a process that should significantly enhance its utility for dynamic studies but that comes at the cost of significantly weaker protein expression[46]. We first engineered a rtTA construct where a PEST proteolytic signal was inserted at its N terminus (PEST-rtTA)[52].

We used the electroporation technique to test this constructs in the dorsal region of the neural tube (Fig. 3c), known to respond to TCF/LEF signaling[35]. In the control experiment, we tested a native (not destabilized) rtTA construct, and we observed that the addition of doxycycline to electroporated embryos led to a strong response, with many visible dTomato[+] neural cells (Fig. 3a, d, e). Expectedly, fusing PEST sequences to the rtTA construct led to a strong decrease in the efficiency of the response, with only few visible dTomato[+] neural cells (Fig. 3a, f, g). To address the reduced expression levels, we incorporated to the rtTA/PEST construct translation enhancer sequences (IVS/Syn21/p10; see above). The addition of translational enhancers to the rtTA-PEST sequence restored a labeling efficiency that was comparable to that observed with the original rtTA construct (Fig. 3a, h, i). This suggests that the two-step strategy (destabilization/translation enhancement) generated a sensitive Tet-on system that dynamically responds to doxycycline exposure when TCF-LEF/β-catenin dependent signaling is active.

We tested the efficiency of this tracing system. Tracing TCF-LEF/β-catenin-responding cells involves a succession of molecular steps (doxycycline-triggered activation of Cre expression–excision of Stop sequences, and expression of the lineage tracing fluorescent protein, (Fig. 3b). To evaluate the efficiency of the system to permanently label all cells experiencing TCF-LEF/β-catenin dependent signaling, we substituted the 16TF promoter with a CAGGS ubiquitous promoter. This should theoretically lead to the activation of the tracing fluorescent protein in all electroporated cells upon doxycycline addition. The VLL of brachial somites were co-electroporated with this plasmid mix, and embryos were exposed to a doxycycline solution for two consecutive days and then analyzed one day later at E4.5 (Fig. 3j). We observed that 98% of the electroporated cells, labeled by the expression of the constitutive mVenus protein, also expressed dTomato (Fig. 3k–m). This near-perfect correlation between the expression of mVenus and dTomato suggests that, despite significant destabilization of the rtTA protein and the many molecular steps required to activate the reporter, the doxycycline-mediated induction of dTomato fluorescence by the Tet-on/CRE system we designed is highly efficient. Furthermore, this experiment indicates that despite using multiple independent plasmids (five, including the transposase construct), all electroporated cells appear to have simultaneously incorporated them. This efficient and dynamic tool, that we named *TCF-Trace*, is the first molecular tool aimed at following the fate of cells experiencing temporary bursts of TCF-LEF/β-catenin-dependent signaling. We therefore proceeded to investigate the fate of myogenic precursor cells labeled by TCF-Trace.

## Two distinct progenitor populations co-exist in early limb myogenesis

The TCF-Trace system displays accurate temporal labeling of targeted cells without significant lag from previous cell history. This temporal precision is crucial in our experimental design, as we electroporate VLL cells that display a high TCF-LEF/β-catenin-dependent activity at E2.5 (Fig. 1c, d), an activity, likely linked to their epithelial state[53–55], that we do not intend to trace. We therefore initiated the lineage fate of PAX7[+]/MYF5[+] progenitors present in the limb from embryonic day 4.5 (E4.5).

To do this, we electroporated the VLL of E2.5 brachial somites with a combination of the three plasmids described above, together with a ubiquitously expressed mVenus as an electroporation marker and the transposase plasmid. Subsequently, at E4.5, 5.5, and 6.5, doxycycline was added to developing embryos, aiming to label all progenitors activating TCF-LEF/β-catenin-dependent signaling during that time window. We analyzed the embryos at E7.5, at a time when the activity of the 16TF-VNP reporter is almost extinguished (Fig. 1l, Fig. 4a). We tested for expression of PAX7, mVenus, and dTomato proteins (Fig. 4b–d). We observed numerous electroporated (green) myofibers containing dTomato[+] nuclei (red), indicating a massive contribution of the TCF-LEF/β-catenin[+] progenitors to myotube formation (Fig. 4b).

Strikingly, this analysis unveiled an additional finding: a significant proportion (65–72%) of PAX7[+], electroporated progenitors were not labeled by TCF-Trace. This was observed both in the proximal and distal region of the muscle masses (arrows in Fig. 4c, d). As we demonstrated the high sensitivity of the tracing system we designed (Fig. 3j–m), it is unlikely that the absence of label is due to a failure to detect and trace all TCF-LEF/β-catenin[+] cells. Instead, it suggests the presence of a population of progenitors in the developing limb that never activate TCF-LEF/β-catenin signaling, co-existing with the TCF-Trace[+] progenitor population. This intriguing observation triggered further investigation into the long-term fate of these distinct populations.

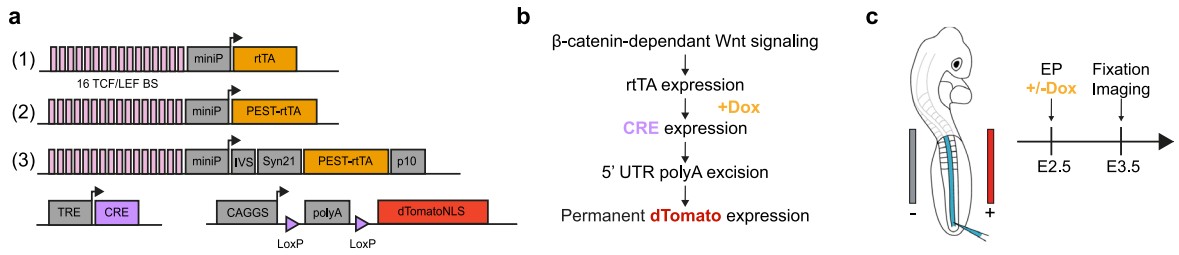

*CAGGS:H2B-TagBFP ; TRE:CRE ; CAGGS:LoxP-polyA-LoxP-dTomatoNLS*

## TCF-LEF/β-catenin⁺ myogenic precursors differentiate into primary myotubes

To streamline myotube analyses, we engineered a second version of TCF-Trace, where a cytoplasmic forms of the fluorescent protein eGFP was used. This enabled straightforward identification of the entire myotube diameters on cross sections. The electroporation tracer was a cytoplasmic form of dTomato driven by the Myosin Light Chain (MLC[56]) promoter, known to be expressed in myofibers. On cross sections at E9.5, we observed that nearly all (93%) electroporated myofibers (in red) were positive for eGFP (in green; Fig. 4e, h). At this

stage in chicken embryos, only primary myotubes are present in muscle masses[10]. Based on this temporal criterion, we therefore hypothesized that the TCF-LEF/β-catenin⁺ population represents the precursors of primary myotubes.

To confirm this, we proceeded with an immunostaining approach. It is widely accepted that all slow myosin-expressing myofibers are primary myotubes[19–22]. As reports on slow myosin expression in chicken are scattered over many publications and developmental stages, we performed a comprehensive survey of slow myosin expression in the chicken forelimb from embryonic to fetal stages of

**Fig. 3 | TCF-Trace, a tool to follow the fate of cells experiencing TCF-LEF/β-catenin dependent signaling. a, b** Schematics of the constructs tested. The three constructs comprise 16 TCF/LEF binding sites upstream of a minimal promoter driving the expression of a rtTA (1) or a rtTA fused with a PEST sequence at its N-terminal part (2) or a rtTA fused with a PEST sequence at its N-terminal part, flanked by translational enhancers (3). All constructs were co-electroporated with a plasmid containing a rtTA-dependent (Tetracycline-Response Element, TRE) CRE recombinase and another plasmid containing a CRE-inducible nuclear dTomato. **b** Upon a response to TCF/LEF signaling in presence of doxycycline cells are permanently labeled with dTomato. **c** Embryos were electroporated in the neural tube at E2.5, induced with doxycycline and analyzed one day later. **d–i** Dorsal view of confocal stacks of E3.5 neural tube (NT) electroporated with a ubiquitous

H2B-TagBFP and the rtTA plasmid (**d, e**), the destabilized rtTA (**f, g**) or the desta-bilized and boosted rtTA (**h, i**). The dotted lines delineate the electroporated, right side of the neural tube. **j** Quantification of (**d–i**), each dot represents an embryo ($n = 4, 4, 4, 5, 4$ and 4 for each condition, respectively). Statistical comparison was performed using the non-parametric Kruskal-Wallis test for multiple comparisons, followed by Dunn-Bonferroni post-hoc test. * = $p < 0.05$. **k, m** Dorsal view of confocal stacks of E4.5 limb buds electroporated at E2.5 with an ubiquitous mVenus, a ubiquitous rtTA, the CRE and dTomato plasmids, doxycycline was added at E2.5 and E3.5. **m** Quantification of the percentage of dTomato cells within the mVenus+ population, each dot represents a limb bud ($n = 6$). Box plots show the median (center line), 25th and 75th percentiles (box limits), and whiskers extending to the smallest and largest values within 1.5 x IQR. Scale bars: 50 μm (**d–i, l**) or 100 μm (**l**).

development (Supplementary Fig. 8), and observed that its expression significantly progresses along development, starting to be visible at E9.5 in a few of the muscle masses present at that stage (Supplementary Fig. 7c), and steadily intensifying to encompass all muscles at E18.5 (Supplementary Fig. 7d–h). From E12.5 until E18.5, slow (primary) myofibers are progressively surrounded by small, fast (presumably secondary) myofibers (Supplementary Fig. 8i–p). To investigate the slow vs fast signature of the TCF-LEF/β-catenin-derived myofibers during embryonic and fetal development, we performed a lineage tracing experiment similar to the one above but analyzed the embryos at stage E16.5 (Fig. 4i–j). We observed that nearly all (98%) slow myosin-positive myotubes expressing the electroporation marker derived from the TCF-LEF/β-catenin+-derived lineage (Fig. 4i, j). Therefore, based on temporal and biochemical criteria, these experiments demonstrate that TCF-LEF/β-catenin+ progenitors present in the growing limb bud constitute the cellular origin of primary myotubes.

### TCF-LEF/β-catenin- myogenic precursors differentiate into secondary myotubes and satellite cells

We then wondered what is the origin of secondary myotubes. Secondary myotubes have been shown to appear after E9 in chicken, often surrounding primary myotubes (Supplementary Fig. 8i,j)[10,57]. We followed the fate of TCF-Trace-positive and -negative progenitors, using the same protocol as in the previous experiment set, but left the embryos to develop until E12.5. In contrast to the analyses done at E9.5, we found that 21% of electroporated (red) myotubes were not labeled by the TCF-Trace system (green Fig. 4f–h). Often, we observed that dTomato-only myotubes were smaller than eGFP-positive myotubes and that they were located at their periphery, which are typical characteristics of secondary myotubes (Fig. 4g, arrowheads). Therefore, based on temporal and morphological criteria, these experiments strongly suggest that TCF-LEF/β-catenin- progenitors present in the growing limb bud constitute the cellular origin of secondary myotubes. Interestingly, in a similar analysis done at E16.5, we observed a significant decrease in the proportion of TCF-Trace-negative myotubes, paralleled by an increase in the proportion of TCF-Trace positive myotubes (Fig. 4h). It is possible that the shift in the proportions of TCF-Trace+ and TCF-Trace- myotubes from E12.5 until E16.5 is due to a mixing between both lineages, either through myoblast to myofibers fusion or through fusion between myofibers.

Finally, we determined whether satellite cells originate from one, the other, or both progenitor populations. VLL cells of E2.5 brachial somites were electroporated with a combination of the three plasmids described above, together with a ubiquitously expressed H2B-Achilles as an electroporation marker. At E4.5, E5.5, and E6.5, doxycycline was added to developing embryos. In birds and rodents, satellite cells are the only PAX7+ cells present in muscle masses in late fetal stages, just before hatching/birth[6,58–61]. We explored whether PAX7+ cells present at E16.5 were labelled by the tracing system. We found that the vast majority (87%) of electroporated satellite cells present in limb muscle masses were TCF-Trace-negative, while a minority (13%) were positive (Fig. 4k, l). This finding suggests that most satellite cells present in

muscles at hatching derive from the TCF-LEF/β-catenin- progenitor population present in early limb buds, with a minor contribution originating from the TCF-LEF/β-catenin+ lineage.

### TCF-LEF/β-catenin signaling regulates the spatial distribution of primary progenitors

With the TCF-LEF/β-catenin reporter revealing pivotal connections between the embryonic origins of primary and secondary myotube lineages and adult muscle stem cells, it naturally prompted us to investigate how this signaling pathway influences these critical developmental processes. Canonical, β-catenin-dependent Wnt signaling has been proposed to fulfill various functions during embryonic myogenesis, synergizing or antagonizing with other signaling pathways, such as Sonic Hedgehog, Bone Morphogenic Proteins, or NOTCH[40,62–65]. In the limb, conflicting data suggest that β-catenin-dependent Wnt signaling activates, inhibits or is dispensable for embryonic myogenesis[66–69]. The expression of a large variety of Wnt signaling molecules (e.g. Wnt secreted signals, Wnt receptors and co-receptors, Wnt antagonists) during limb development[70–72] and different experimental procedures may explain those contradictory results. Furthermore, our discovery of the existence of two co-existing populations of limb muscle progenitors, distinguished by their response to TCF-LEF/β-catenin-dependent signaling, necessitates a reevaluation of the role of Wnt signaling in limb myogenesis. The precise spatio-temporal control provided by in vivo electroporation offers a powerful alternative to classical approaches, potentially clarifying the discrepancies observed in previous studies.

To address the role of Wnt β-catenin-dependent signaling during limb myogenesis, we used a dominant-negative form of Lef1 (DN Lef1), known to efficiently inhibit Wnt β-catenin-dependent signaling[73]. This construct was electroporated at E2.5 in the lateral portion of brachial somites to target the limb muscle progenitor population. Its effect was evaluated at E4.5, at a time when the 16TF-VNP reporter is active in 50% of the progenitor population (Fig. 1k, l). We examined myogenic progenitor proliferation and differentiation under this experimental setting. No differences were observed in the progenitor proliferation, as determined by phospho-histone H3 or EdU labeling (Supplementary Fig. 9a–f). Similarly, the early myogenic differentiation program remained unaltered, since the expression of the differentiation marker MYF5 was not altered (Supplementary Fig. 9g–i). To evaluate the role of TCF/LEF signaling in late myogenic commitment, we designed an inducible form of DN-Lef1 by placing its coding sequence, along with that of a nuclear dTomato, under the control of a bidirectional TRE promoter (pBI). We co-electroporated this construct with a ubiquitous rtTA-expressing plasmid and added doxycycline at E4.5, once myoblasts have migrated into the limb bud but have not yet expressed the terminal differentiation marker *MyoG*. Two days after doxycycline induction (E6.5), the percentage of MYOG+ cells was similar to that of controls (Supplementary Fig. 9j–m).

Interestingly, the spatial distribution of progenitors electroporated with the ubiquitous DN Lef1 revealed a notable effect. At E4.5, a subset of these cells exhibited a tendency to remain in a proximal

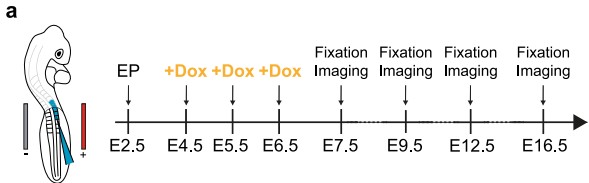

*CAGGS:mVenus; 16xTF:IVS-Syn21-PESTrtTA-p10 ; TRE:CRE ; CAGGS:LoxP-polyA-LoxP-dTomatoNLS*

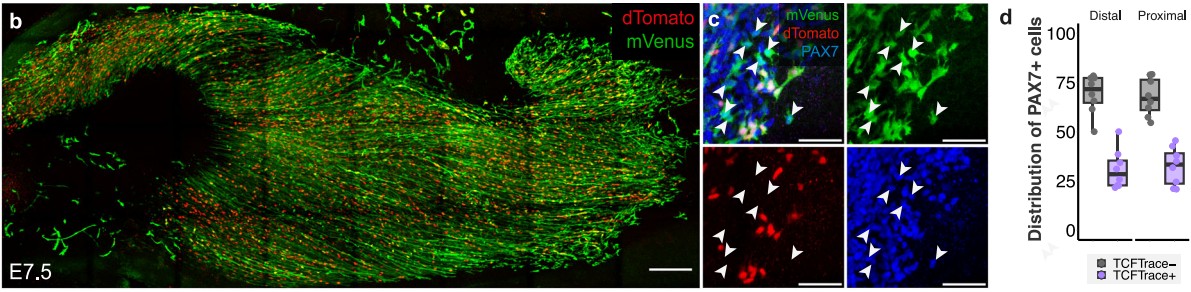

*MLC:dTomato ; 16xTF:IVS-Syn21-PESTrtTA-p10 ; TRE:CRE ; CAGGS:LoxP-polyA-LoxP-eGFP*

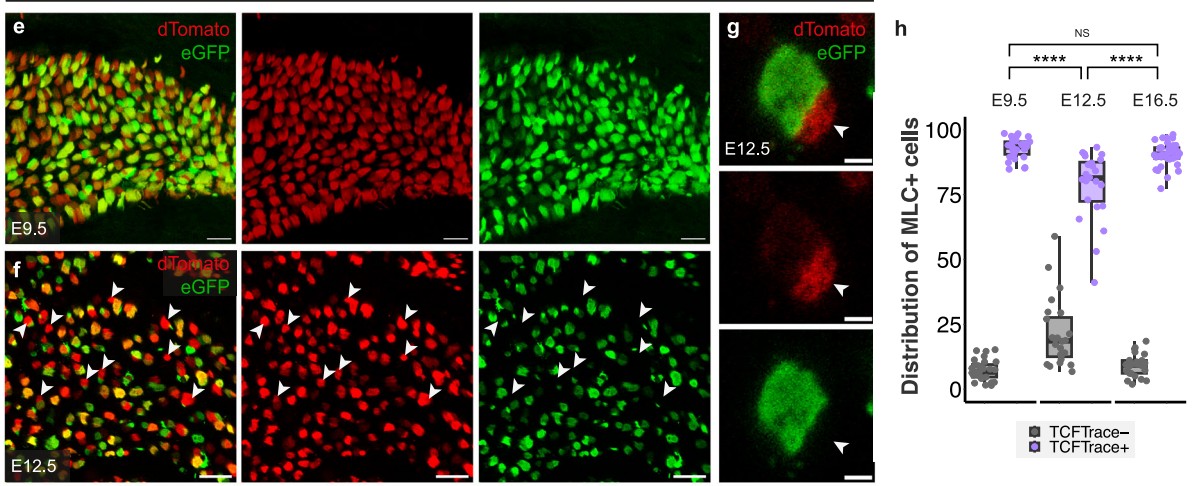

*MLC:dTomato ; 16xTF:IVS-Syn21-PESTrtTA-p10 ; TRE:CRE ; CAGGS:LoxP-polyA-LoxP-eGFP*

*CAGGS:H2BAchilles; 16xTF:IVS-Syn21-PESTrtTA-p10 ; TRE:CRE ; CAGGS:LoxP-polyA-LoxP-dTomatoNLS*

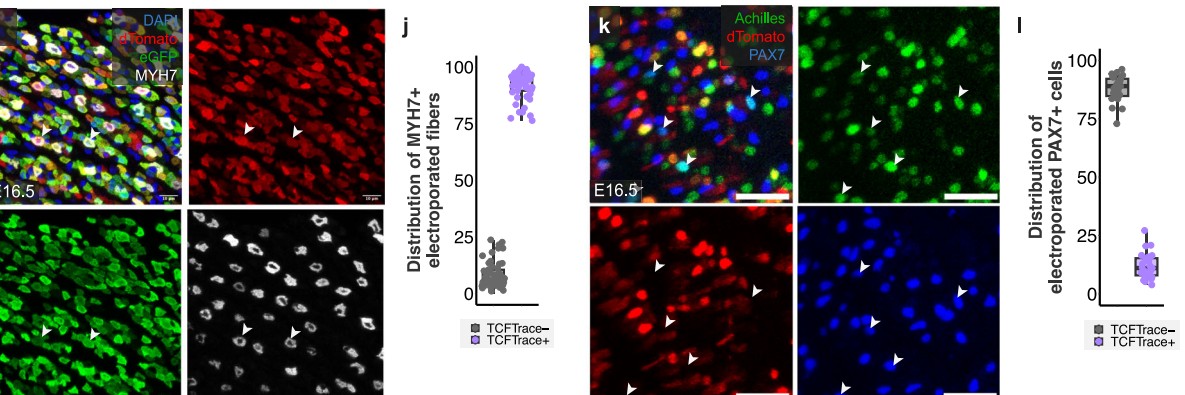

location, closer to the somites from which they originated. In contrast, those that migrated distally showed an increased dispersion away from the limb axis. Quantitative analysis confirmed a significant difference in distribution between the progenitors expressing DN Lef1 and the controls (Fig. 5a–d). To rule out the possibility that this phenotype was due to the inhibition of TCF-LEF/β-catenin signaling in the VLL, where it likely maintains its epithelialization state[53–55], we repeated the experiment with the inducible form of DNLEF1 construct by activating its expression at E4.5 and examined embryos at E6.5. The results were consistent with the previous experiment (Supplementary Fig. 9n, o), reinforcing the role of TCF-LEF/β-catenin signaling in the control of muscle progenitor distribution.

While the many Wnt ligands expressed during limb formation in amniote embryos could potentially regulate the migration and/or the

**Fig. 4 | Lineage analysis of TCF-Trace⁺ and TCF-Trace⁻ populations. a** TCF-Trace was induced with doxycycline during the time period (E4.5-E6.5) in which muscle progenitors respond to TCF/LEF signaling. Brachial somites of E2.5 embryos were electroporated, TCF-Trace was induced 2–4 days later, and embryos were analyzed at various stages from E7.5 to E16.5. **b, c** Dorsal view of confocal stacks of E7.5 limb bud electroporated with a ubiquitous mVenus, the TCF-Trace lineage tool driving the expression of a nuclear dTomato and stained for PAX7. **c** is an enlargement of (**b**). Arrowheads in (**c**) indicate electroporated TCF-Trace⁻ muscle progenitors **d** Quantification of the percentage of TCF-Trace⁻ or TCF-Trace⁺ cells in the PAX7⁺ electroporated population at E7.5 in the proximal and the distal part of the muscle mass, each dot represent a limb bud. **e, f** Transverse sections of E9.5 (**e**) and E12.5 (**f**) limb buds electroporated with a myofiber-specific dTomato and the TCF-Trace lineage tool driving the expression of eGFP. Arrowheads in (**f**) indicate electroporated TCF-Trace⁻ myofibers. **g** Representative example of a TCF-Trace⁺ myotube surrounded by a smaller TCF-Trace⁻ myotube (white arrowhead). **h** Quantification of the percentage of TCF-Trace⁻ and TCF-Trace⁺ myotubes in the MLC⁺ electroporated population, each dot represents a section, *n* = 7 limbs for each condition, each dot represents a section, *n* = 28, 24, and 24 for E9.5, E12.5, and E16.5,

respectively. Statistical comparison was performed using the non-parametric, Kruskal-Wallis test for multiple comparisons, followed by Dunn-Bonferroni post-hoc test. NS: *p* > 0.05, ****: *p* < 0.0001. **i** Transverse sections of E16.5 limb buds electroporated with a myofiber-specific dTomato, the TCF-Trace lineage tool driving the expression of eGFP, and immunostained for MYH7. Arrowheads indicate electroporated TCF-Trace⁺ myofibers expressing MYH7. **j** Quantification of the percentage of TCF-Trace⁻ or TCF-Trace⁺ myotubes in the MYH7⁺ electroporated myotubes, each dot represents a section, *n* = 6 limbs each dot represents a section (*n* = 52 sections). **k** Transverse sections of E16.5 limb buds electroporated with a nuclear Achilles, the TCF-Trace lineage tool driving the expression of a nuclear dTomato and stained for PAX7. Arrowheads indicate electroporated TCF-Trace⁻ muscle progenitors. **l** Quantification of the percentage of TCF-Trace⁻ or TCF-Trace⁺ cells in the PAX7⁺ electroporated population, each dot represents a section, *n* = 7 limbs, each dot represents a section (*n* = 25 sections). Box plots show the median (center line), 25th and 75th percentiles (box limits), and whiskers extending to the smallest and largest values within 1.5xIQR. Scale bars: 10 μm (**g, i, k**), 20 μm (**e, f**), 50 μm (**c**), or 200 μm (**b**).

distribution of the primary myotube progenitors, an important question was whether Wnt signaling controls this process directly, acting as a trophic factor, or indirectly, through transcriptional regulation of another molecule. To test this, we injected *Wnt1*-expressing cells into early limb buds at E3.5 and analyzed the distribution of (PAX7⁺) muscle progenitor cells one day after (Supplementary Fig. 10a–d). These manipulations did not lead to detectable defects in the normal distribution of muscle progenitors in experimental embryos (Supplementary Fig. 10e–j), suggesting that, while Wnt signaling regulates progenitor migration, Wnt ligands do not act as positional cues in this process.

## A common developmental program underlies chicken and human limb myogenesis

A multitude of genes and signaling pathways are known to regulate muscle progenitor migration during embryogenesis[74–77], making a candidate gene approach to test targets of TCF-LEF/β-catenin signaling in primary myotube progenitors a tedious endeavor. Therefore, we adopted a transcriptomic approach, performed on limb muscle progenitors co-electroporated with a ubiquitously-expressed nuclear dTomato and the 16TF-VNP reporter, followed by cell sorting based on red fluorescence (Supplementary Fig. 11). This enriched population of myoblasts was then subjected to single-cell RNA sequencing (scRNA-seq) analysis (Fig. 5e).

Using unsupervised graph-based clustering combined with FFT-accelerated interpolation-based t-SNE (FIt-SNE), we identified several distinct clusters of cells (Fig. 5e). We employed negative selection to exclude non-muscle contaminating cells, identified by specific markers (e.g., *Csf1r* for macrophages; *Flt1* for endothelial cells, see Supplementary Fig. 12a), while retaining cells expressing *Pax7*, a known marker of muscle progenitors. Within *Pax7*⁺ cells, we identified six distinct clusters that, based on the expression of known factors (Supplementary Fig. 12b, c, Supplementary Data 1), represent stages of proliferation and differentiation of muscle progenitors (Fig. 5f). By mapping for the presence of transcripts from the two electroporated transgenes within these clusters, individual cells could be grouped into two distinct populations: 16TF-VNP⁻ (corresponding to secondary myotube and satellite cell progenitors) and 16TF-VNP⁺ (encompassing progenitors of the primary myotube lineage; red and yellow dots, respectively, Fig. 5g, h). The 16TF-VNP⁺ cells were absent in Cluster 5, which comprises progenitors in the late stages of myogenic differentiation (*e.g.*, *MyoD*⁺, *MyoG*⁺, *Tmem8c/Mymk*⁺; Supplementary Fig. 12c), consistent with earlier data (Fig. 2). However, cells from the two populations were found in all other clusters, suggesting that at that stage of development, their transcriptional signatures are largely similar.

Despite this high similarity, we identified 35 genes differentially expressed in the 16TF-VNP⁺ population compared to the 16TF-VNP⁻ cells (Supplementary Fig. 12d, Supplementary Data 2). This analysis revealed that i) known members and targets of Wnt canonical signaling were expressed at a higher level in the 16TF-VNP⁺ cell population, while secreted inhibitors of the pathway were more prevalent in the 16TF-VNP⁻ cell population (Supplementary Fig. 12d, e); ii) factors known to regulate cell migration in various contexts were overexpressed in the 16TF-VNP⁺ population (e.g. *Cxcr4* during myogenesis[78]; *Epha7* and *Nectin3* in the hematopoietic system[79,80]; *Epha7* in the nervous system[81,82]; *Cadm1, Wasf3, Sh3pxd2a, Mylk, Brca1, Prrx1* and *Erc1* in cancer[83–89]; Fig. 5i). To further analyze our findings, we compared our dataset with recently published single-cell RNA sequencing data from human embryonic limbs[90]. Mapping myogenic cells between 5 to 9 weeks of development, this study had identified two distinct myogenic trajectories, termed first (or "embryonic") and second (or "fetal") myogenesis and a transient intermediate state linking the two (Supplementary Fig. 13a-d). Strikingly, several genes enriched in the 16TF-VNP⁺ population (*CADM1, NECTIN3, CXCR4,* and *LEF1*) were also highly expressed during human embryonic myogenesis (Supplementary Fig. 13e). For each human myogenic cell, we calculated a score based on the level of expression of the 35 most highly expressed genes in the 16TF-VNP⁺ population in chicken (see methods for a detailed explanation). Cells with the highest score preferentially mapped to the first "embryonic" and transitory state of myogenesis of the human dataset (Supplementary Fig. 13f). Intrigued, we asked whether Wnt-TCF/LEF signaling might also characterize embryonic myogenesis in human. To test this, we calculated a score for each human myogenic cell, based on genes known to be Wnt targets during muscle regeneration[91,92]. This analysis revealed a strong correlation between the embryonic trajectory and high Wnt target gene expression (Supplementary Fig. 13g), suggesting a conserved association between Wnt signaling and primary myogenesis in both chicken and human.

Together, the strong transcriptomic similarities between TCF/LEF-responsive, primary myotube lineage progenitor cells in chicken and the 'embryonic' myogenic population identified in human limbs, combined with their shared association with active Wnt signaling, support the idea that these populations are homologous, pointing to a conserved developmental program between birds and mammals.

## CXCR4 acts downstream of TCF-LEF/β-catenin to regulate the spatial distribution of primary progenitors

In the transcriptomic analyses described above, the chemokine receptor *Cxcr4* emerged as a particularly intriguing candidate. In mouse and chicken embryos, *Cxcr4*, together with its cognate ligand *Sdf1*, regulates muscle progenitor migration. *Cxcr4* is expressed in a

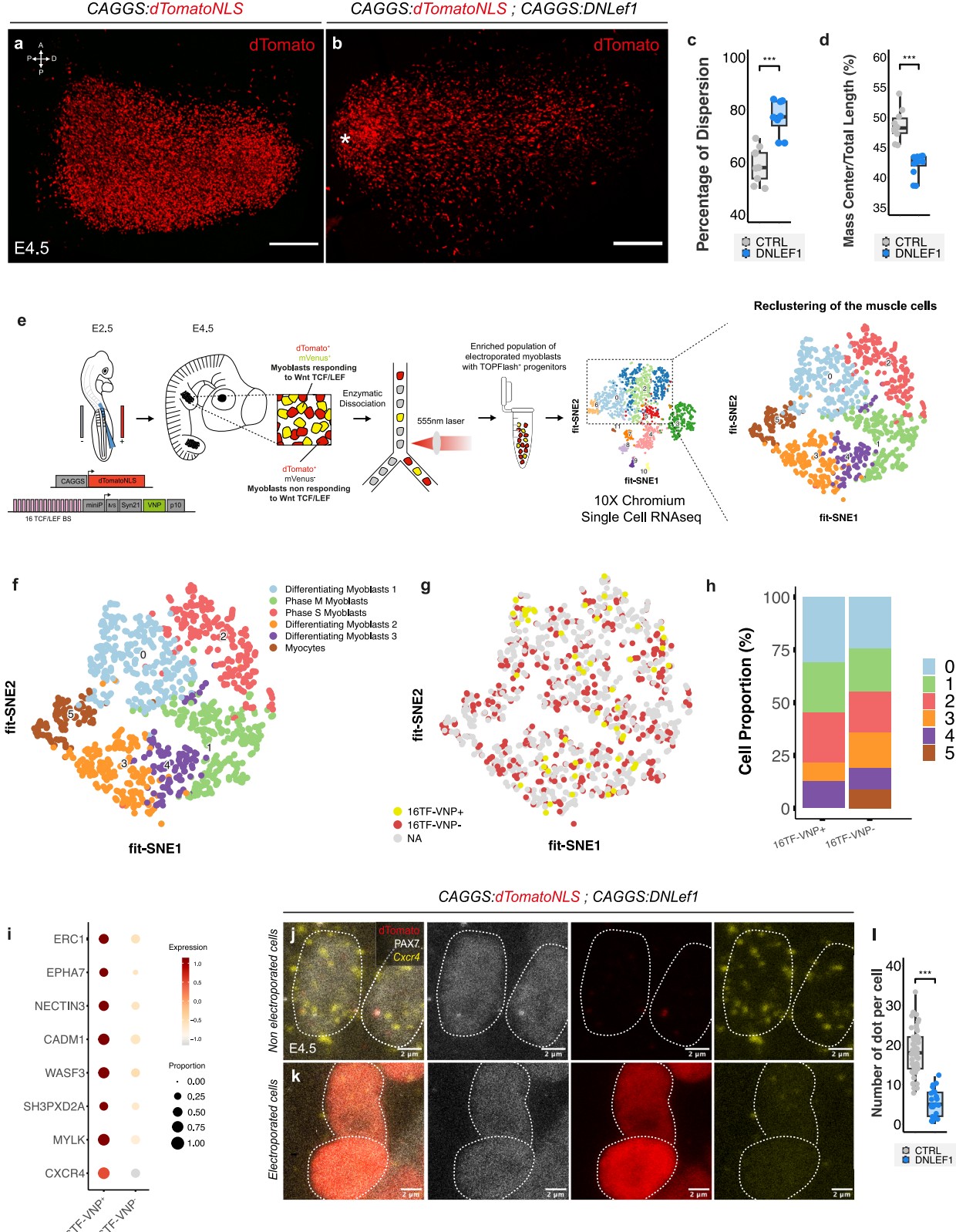

subset of limb muscle progenitors, while *Sdf1*, expressed in the limb mesenchyme, acts as a trophic factor to guide the migration of *Cxcr4*-positive progenitors. Interestingly, as was shown for TCF-LEF/β-catenin-positive cells (Fig. 2b–g), *Cxcr4* and *MyoD* expressions are mutually exclusive[78].

These intriguing parallels suggested that *Cxcr4* might be a target of TCF-LEF/β-catenin-dependent signaling in early limb progenitors of the primary myotube lineage. To test this hypothesis, we electroporated the DN Lef1 construct at E2.5 in the VLL of brachial somites. Two days later, the limb buds were analyzed by HCR FISH[93] to detect the expression of *Cxcr4*. We observed that the expression of *Cxcr4* was strongly downregulated in electroporated PAX7+ myoblasts compared to the PAX7+ WT myoblasts (Fig. 5j–l). Altogether, these results strongly suggest that TCF-LEF/β-catenin-dependent signaling in the

**Fig. 5 | TCF-LEF/β-catenin controls the spatial distribution of primary myotube progenitors via *Cxcr4*. a**, **b** Dorsal view of a confocal stack of limbs electroporated with a ubiquitously expressed nuclear dTomato alone (**a**) or together with DN Lef1 (**b**). The asterisk in (**b**) indicates the clump of cell that remains located proximally. **c**, **d** Quantifications of the repartition of muscle progenitors in the limb, each dot represents an embryo (*n* = 9 and 8 for (**c**) and *n* = 10 and 12 for (**d**)). Statistical comparison was performed using the two-sided non-parametric Wilcoxon test. *** = *p* < 0.001. **e** Procedure followed to perform scRNAseq on electroporated limb muscle progenitors. **f** Six distinct clusters were identified within the total

population of myoblasts. **g**, **h** 16TF-VNP⁻ (red) and 16TF-VNP⁺ (yellow) distribute in all, but cluster 5. **i** List of migration-associated genes differentially expressed in the 16TF-VNP⁺ and 16TF-VNP⁻ populations of myoblasts. **j**, **k** HCR FISH to detect *Cxcr4* expression in control limb myoblasts (**j**) or after DN Lef1 over-expression (**k**). **l** Quantification of (**j**, **k**), 4 limb buds were analyzed and each dot represent a cell (*n* = 75 cells in total). Statistical comparison was performed using the two-sided non-parametric Wilcoxon test. *** = *p* < 0.001. Box plots show the median (center line), 25th and 75th percentiles (box limits), and whiskers extending to the smallest and largest values within 1.5xIQR. Scale bars: 100 μm (**a**, **b**).

progenitors of the primary myotube lineage is essential to regulate their distribution in the growing limb bud, partially or entirely through the transcriptional regulation of the *Cxcr4* receptor. Further studies will elucidate whether Wnt signaling regulates the other migration regulators we identified, which could have broader implications for the spatial organization and function of primary myotubes.

## Discussion

The discoveries outlined in this study address two enduring puzzles that have persisted within the field of myogenesis for four decades.

i) Firstly, it resolves the longstanding question of the developmental origin of primary and secondary myotubes. It unequivocally demonstrates that these myotubes arise from two distinct myogenic progenitor populations that coexist early in the limb and are distinguished by their TCF-LEF/β-catenin-dependent Wnt signaling activity. While a recent transcriptomic analysis of the early developing human limb identified two co-existing myogenic lineages, their long-term developmental fates had not been explored[90]. The striking similarities we identified between human and chicken transcriptomic signatures, combined with our long-term lineage analyses, strongly suggest that the ontogenesis of primary and secondary myogenesis is governed by conserved molecular and cellular mechanisms across all amniotes.

ii) Secondly, our finding clarifies a controversy surrounding the lineage association between embryonic and fetal myoblasts. The observation that myogenic precursors isolated from embryonic and fetal limbs exclusively generated primary or secondary myofibers, respectively, had led to the conclusion that myogenic progenitors consist of a homogeneous precursor population that evolves as development progresses[26,28,29]. Contradictory observations showed that embryonic and fetal myoblasts are two distinct precursor populations, sequentially migrating into the limb mesenchyme[32], or present from the onset of limb formation[10,34]. Our study resolves this controversy by demonstrating that the myogenic progenitors present during limb development comprise a mixed population of precursors for primary and secondary myotubes. A surprising outcome of our findings is the identification of a shared origin for limb secondary myotubes and satellite cells. Although it is long established that all limb muscles, including its satellite cell component, originate from the VLL[94,95], the developmental path that VLL cells follow to generate satellite cells, once they have migrated into the limb mesenchyme, has not been investigated.

These findings prompt a re-evaluation of muscle development in both trunk and limb. Trunk muscle formation has been thoroughly studied and is understood to occur in two distinct stages: i) an initial stage, where epithelial cells from the edge of the dermomyotome closest to the neural tube contribute to the formation of the primary myotome, composed of mononucleated myocytes[5,7,96]. This primary myotome then transitions into a polynucleated structure through fusion with cells from other dermomyotome borders[36]; ii) a second stage, where satellite cell progenitors emerge from the central dermomyotome and migrate into the primary myotome. These progenitors initially fuse with each other to form additional polynucleated fibers, with a subset set aside to become satellite cells[6,58,59].

The similarities between these processes in the trunk and our observations in the limb suggest a compelling hypothesis: despite the apparent differences in their origins—trunk muscles forming within the structured somite and limb muscles arising from migrating progenitors—both may follow a similar developmental pattern involving two distinct progenitor populations with different fates: i) the primary myotome and cells from the dermomyotome borders in the trunk may be analogous to the primary myotube lineage we identified in the limb; ii) cells from the central dermomyotome could correspond to the TCF-Trace-negative progenitors in the limb, along with their derivatives, the satellite cells. Further analyses will be necessary to determine whether the biochemical and cellular characteristics of these myogenic progenitor populations support a unified model of trunk and limb muscle formation.

As was observed in the trunk[36], it is fascinating that limb muscle progenitors carefully choose their fusion partners in an environment where all progenitors and myofibers are tightly intertwined, such that TCF-Trace⁺ progenitors only fuse to myofibers derived from their own lineage, while TCF-Trace⁻ progenitors, at least initially, fuse to myofibers from their own lineage. The cellular and molecular underpinning of such intricate fusion pattern remains to be uncovered.

Lastly, the use of the TCF-LEF/β-catenin-dependent signaling reporter was pivotal in revealing the embryonic origins of two distinct populations of myogenic progenitors and their derivatives. This prompted a deeper investigation into the role of this pathway in limb myogenesis, leading to the discovery of its involvement in the spatial distribution of primary myotube progenitors. The identification of *Cxcr4*, a known regulator of limb progenitor migration, as a transcriptional target of TCF-LEF/β-catenin-dependent signaling underscores Wnt canonical signaling's role in migration, likely acting indirectly through *Cxcr4*. Interestingly, *Cxcr4* loss of function was shown to result in myoblast apoptosis[78], suggesting that while Wnt signaling may not directly influence cell survival or proliferation, it could act upstream of genes like *Cxcr4* that do so. Finally, the identification of additional transcriptional targets associated with cell migration across various contexts suggests that *Cxcr4* may only partially control myoblast migration in the limb.

This finding adds yet another layer to the diverse functions of TCF-LEF/β-catenin-dependent signaling during myogenesis, complementing the extensive body of research on Wnt signaling in this process (see refs. 62,63,97 for reviews). While many studies have suggested an essential role for Wnt signaling in the cell decision-making to initiate myogenesis in somites, the intricate morphogenetic processes that accompany this cell fate decision—such as the epithelial-mesenchymal transition of dermomyotomal cells, their migration into the primary myotome, and their orientation and growth along the antero-posterior axis—have made it challenging to pinpoint the specific roles of canonical Wnt signaling. This complexity might explain the contradictory findings in the field[3,64,65,98–100].

Divergent data also exist regarding Wnt signaling's role in limb myogenesis and adult muscle regeneration, with studies suggesting that Wnts can promote, inhibit, or be dispensable for myogenesis[67,101,102]. Given the extensive literature on Wnts' potential roles in myogenesis and the numerous Wnts expressed during limb development[71,103,104], it was surprising to observe that the 16TF-VNP

reporter activity was restricted to a relatively short time window of limb myogenesis. It was completely absent in one lineage and silent during the majority of late myogenesis, including in satellite cells. These observations rule out the possibility that Wnts, via canonical, TCF-LEF/β-catenin-dependent signaling, play a role in late myogenesis, tissue patterning, or the emergence of the satellite cell population.

Our results align with previous studies in mice and chicken, that indicated that Wnt signaling is dispensable for *Myf5* and *MyoD* expression and myofiber formation in early myogenesis[53,69]. Hutcheson et al. also suggested that there exists a PAX7+ lineage that contributes to late myotube formation. However, this study did not analyze the significance of those findings the context of the primary and secondary myotube lineages.

The discovery of a molecular signature distinguishing primary from secondary myotube lineages is poised to revitalize a field that has been largely dormant for years due to the absence of specific molecular markers for these lineages. This breakthrough opens new avenues for investigation, enabling comprehensive molecular characterizations of both lineages. This will allow addressing critical questions about the developmental pathways their progenitors follow to generate functional muscles in the embryo and their roles during development, as well as during regeneration and repair in the adult. The 16TF-VNP reporter and the TCF-Trace system, combined with advanced "omics" technologies, will be instrumental in deciphering the molecular pathways involved in some of the most enigmatic events of late myogenesis.

## Methods

### In ovo chicken embryos electroporation

Fertilized chicken eggs were obtained from a local breeder. To target wing muscle progenitors, the ventro-lateral lip (referred to as the VLL) of brachial somites in E2.5 chicken embryos was electroporated as described previously[36]. Briefly, chicken embryos were incubated at 37.5 °C until stage HH16 (52 h or E2.5). The plasmid solution was injected into the brachial somites (somites 16−21) using a glass capillary. Three pulses of 50 V, 10 ms in duration, and spaced by 10 ms, were applied directly to the embryo using tungsten and platinum electrodes. Eggs were then sealed and placed back in the incubator until the desired stage was reached. Cells emanating from the VLL and migrating into the wing bud mesenchyme constitute a mixed population of very early migrating endothelial progenitors, followed by myogenic progenitors[38,94,105−107]. This migration occurs over a short window of time, beginning soon after somite formation and lasting 15−20 h[94,105]. We developed an electroporation protocol that primarily targets myogenic progenitors (Supplementary Fig. 2a).

### Expression constructs

All constructs used in this study contained two Tol2 sequences (T2: transposable elements from medaka fish[37]) surrounding the entire construct to allow their stable integration in the chicken genome in the presence of transposase, provided by a co-electroporated transposase plasmid (CAGGS-Transposase). This plasmid does not contain Tol2 sequences and is thus gradually diluted along with cell division.

Depending on the experimental or immunostaining strategies, several plasmids were used to visualize electroporated cells. These plasmids contained the coding sequences for various fluorescent proteins with different spectral characteristics and/or directed to different cellular compartments. The T2-CAGGS:BFP and T2-CAGGS:mVenus plasmids were used to label the cytoplasm of all electroporated cells with blue and green fluorescent proteins, respectively. The T2-CAGGS:H2B-BFP, T2-CAGGS:H2B-Achilles and T2-CAGGS:dTomatoNLS were used to labelled the nuclei of all electroporated cells with blue, green and red fluorescent proteins, respectively. H2B- and NLS-fused fluorescent proteins were used to easily detect single cells.

Wnt binding to its receptor can trigger three downstream pathways. The best characterized (and referred to as "canonical") Wnt cellular response results in the inhibition of the β-catenin destruction complex, ultimately leading to the translocation of β-catenin into the nucleus where it partners with members of the TCF/LEF family of transcription factors to activate various Wnt target genes. To monitor this canonical transcriptional response, efficient transcription-based reporter systems that combine the DNA binding sites of TCF/LEF upstream of a minimal promoter and a reporter gene have been designed, the first of which named TOPFlash[108]. Since the first description of the TOPFlash reporter, many derivatives have been generated that combined multiple TCF/LEF binding sites driving a variety of reporter genes. The high stability of the most commonly used reporters (b-galactosidase, GFP, RFP…) leads to their considerable accumulation in cells, activating the pathway, thus greatly facilitating their detection. However, two major drawbacks are: i) an important lag-time between the activation of the pathway and the accumulation of sufficient amounts of the reporter to be detected, and ii) conversely, the detection of signals in tissues where the activity of the pathway may have already ceased. Most of these constructs are therefore unsuitable to detect rapid spatiotemporal changes in TCF-LEF/β-catenin dependent signaling. We have used here a recently developed reporter of Wnt/β-catenin dependent signaling activity, named 16TF-VNP[35], where the combination of 16 TCF/LEF-binding sites[108], translational enhancers[46,109], and a destabilized (half-life, 1.8 h[110]), fast maturing mVenus fluorescent protein[111] generated the most sensitive to date, yet dynamic, molecular tool to detect TCF/β-catenin-responding cells and tissues in developing embryos, with no detectable background[35,112].

For short-term pseudo-lineage of TCF/LEF responding cells, we inserted a P2A sequence downstream of and in-frame with the 16TF-VNP reporter, followed by a stable form of nuclear mCherry (half-life, about 18 h). This results in the production of two fluorescent proteins with different half-life. This strategy was described previously in *Drosophilia*[46] to generate a dual-color Trans Timer construct that provides spatio-temporal information on signaling pathway activities.

Even stable fluorescent proteins, such as GFP or RFP, are not suitable for long-term studies. For this, we have engineered a tri-partite lineage tracing system based on the Tet-On and the Cre-Lox technologies. A plasmid containing the destabilized rtTA transactivator gene under the control of the 16TF promoter was mixed with a plasmid encompassing a TRE (Tetracycline Response Element) promoter driving the expression of a CRE recombinase (T2-TRE:CRE) and a plasmid coding for a CRE-inducible nuclear dTomato (T2-CAGGS:LoxP-STOP-LoxP-dTomatoNLS) or a cytoplasmic eGFP (T2-CAGGS:LoxP-STOP-LoxP-eGFP).

For blocking the transcriptional activity of the TCF/LEF signaling we used a dominant negative form of the LEF1 protein, consisting of a truncated form of LEF1 that binds the DNA but not to β-catenin, thereby blocking intrinsic LEF1 activity as a dominant negative mutant[113]. This mutant was cloned downstream of the strong and ubiquitous CAGGS promoter (T2-CAGGS:DNLef1) this construction was tested in ref. 35. For testing specifically the latter effect on *MyoG* expression, we cloned this mutant into a bi-directional tetracycline response element that regulates the expression of both a nuclear dTomato and the DNLef1 (T2-pBI:DNLef1/dTomatoNLS). Upon co-electroporation with a ubiquitous rtTA plasmid (T2-CAGGS:rtTA) and addition of doxycycline, this plasmid allows the temporal control of DNLEF1 expression.

As the electroporation technique results in the mosaic expression of transgenes in targeted cells, a ubiquitously expressed electroporation marker was added to identify and analyze all electroporated cells.

To label grafted cells, we used a plasmid coding for a membrane-bound dTomato (T2-CAGGS:mbdTomato). To induce the expression of Wnt1 in the grafted cells, we transfected them with a plasmid

containing the coding sequence of the mouse Wnt1 gene in which a Myc-tag was inserted, under the control of the CMV promoter. This Myc-tagged *Wnt1* construct was shown to induce a duplication of axis in Xenopus embryo as "canonical" TCF/LEF activating WNT proteins do[114].

## Wholemount immunohistochemistry

E3 to E7.5 electroporated limb buds were imaged in whole mount as described[36]. Briefly, after dissection, samples were fixed for 1 h in 4% formaldehyde/PBS at room temperature (RT), briefly rinsed with PBS, pre-incubated in washing buffer (0.2% BSA, 0.1% Triton X-100, 0.2% SDS in PBS) for 1 h at RT, and incubated overnight at 4 °C with the primary antibodies. They were washed at least 5 times during the following day and incubated with secondary antibodies overnight at 4 °C. As fixation significantly decreases the brightness of fluorescent proteins (particularly GFP), we often used primary antibodies against fluorescent proteins to facilitate their detection. The following antibodies were used: rabbit polyclonal antibody (IgG) against dTomato (ab62341, Abcam, 1/1000); chicken polyclonal antibody (IgY) against eGFP, mVenus and Achilles (A10262, Invitrogen, 1/1000); mouse monoclonal antibody (IgG2a) against eGFP, mVenus and Achilles (A11120, Invitrogen, 1/1000); rabbit polyclonal antibody (IgG) against TagBFP (AB233, Evrogen, 1/500); mouse monoclonal antibody (IgG1) against PAX7 (AB528428, DSHB, 1/10); mouse monoclonal antibody (IgG2b) against Myosin Heavy Chain (MF20, AB2147781, DSHB, 1/10) and mouse monoclonal antibody (IgA) against the slow myosin MYH7B (S58, AB528377, DSHB, 1/10). These primary antibodies were used in combination with species-matched secondary antibodies conjugated with Alexa fluorochromes (488 nm, 555 nm, or 647 nm) from ThermoFisher or SouthernBiotech at a concentration of 1/500. After staining, samples were incubated in 50% glycerol/PBS for 1 h at RT and in 80% glycerol/PBS for several hours before analysis.

## Cryosectioning

E9.5 to E18.5 limb buds were fixed for whole-mount imaging as described in the whole-mount imaging section, washed in PBS for 1 hour, and then incubated in 7.5% sucrose/PBS and 15% sucrose/PBS at room temperature (RT) for 30 min to several hours, depending on their size. Next, they were immersed in a pre-melted solution of 15% sucrose and 7.5% bovine skin gelatin at 42 °C with agitation overnight before being placed into cryosectioning molds. The samples were then immersed in dry ice-cold 100% ethanol and stored at −80 °C until sectioning. They were cryosectioned using a Leica cryostat at a thickness of 18 μm. Sections were stored at −80 °C until analysis. Immunohistochemistry was performed as described for whole-mount imaging, using the same set of antibodies, with the exception that secondary antibodies were incubated for only 2 h at RT. Nuclei were detected using DAPI. The slides were mounted in Fluoromount ™. Transversal and longitudinal sections were performed for the analyses of the 16xTF-VNP reporter at E9.5, while only transversal sections of the extensor muscle of the zeugopod were done at E12.5, E14.5, E16.5 and E18.5. Analysis of the TCF-Trace experiments after E9.5 was performed with transversal sections targeting the extensor muscle of the zeugopod.

## Doxycycline induction

Stock solution of doxycycline at a concentration of 20 mg/ml in ddH2O was prepared in advance and stored at −20 °C. A solution at 3.5 μg/ml was prepared by diluting the stock solution into sterile Ringer's solution on the day of the injection; 300 μl of the solution was added per embryo.

## EdU incorporation

50 μl of a 10 mM of EdU solution was added directly onto the embryo that was placed back in the incubator for 1 h. Embryos were then dissected, fixed and immunostained as described above. Once immunostained, samples were pre-incubated in 250 μl of PBS with 1 μl of Alexa fluorophore for 1 h at RT. Separately 150 μl of PBS was mixed with 100 μl of ascorbic acid at 0.5 M and 2 μl of a 1 M CuSO4 solution and added to the pre-incubated samples. Embryos were incubated overnight at 4 °C with agitation, washed at least five times the following day and cleared into glycerol as described.

## *Cxcr4* fluorescent in situ hybridization

The *Cxcr4* probe solution is composed of 21 small probes directly designed by Molecular Instrument and already tested in ref. 115. For a detailed protocol of wholemount HCR on chicken embryos see ref. 115. Electroporated limb buds were collected two days after electroporation, fixed in 4% paraformaldehyde for 1 h at RT. Embryos were washed twice in PBS and dehydrated in growing concentration of MetOH/PBS (25%/50%/75%/100%). Embryos were stored at least one night at −20 °C before being rehydrated in decreasing concentration of MetOH/PBS (75%/50%/25%/PBS) and post fixed for 20 min in 4% PFA for 20 min at RT. Limb buds were then washed twice in PBS for 5 min, then once in a 1:1 solution of PBS/SSCT and finally once in SSCT. The embryos were pre-hybridized in 500 μL of hybridization buffer for 30 min at 37 °C (HCR™ Buffers, Molecular Instruments). The pre-hybridization solution was replaced by the solution with the probes (2 pmol in 500 μL hybridization buffer) and incubated overnight at 37 °C with shaking. After removal of the probe solution, samples were washed 4 times in 1 mL of washing buffer (HCR™ Buffers, Molecular Instruments) for 15 min at 37 °C with shaking. Two 5 min washes in SSCT solution were made at RT. Next, the embryos were incubated in 500 μL of amplification buffer for 5 min at RT (HCR™ Buffers, Molecular Instruments). The hairpins h1 and h2 (30 pmol (10 μL for 500 μL of buffer)) were heated separately at 95 °C for 90 s, left at RT for 30 min minimum in the dark before being added in 500 μL of amplification buffer. The amplifier in the amplification buffer was coupled with Alexa 647 for fluorescent detection. The pre-amplification solution was replaced by the solution containing the amplifiers and incubated overnight in the dark at RT with gentle agitation. Amplifier solution was removed, and embryos were washed in SSCT at RT (2 × 5 min, 2 × 30 min, 1 × 5 min). Embryos were then stained as described above for the dTomato and PAX7, cleared in glycerol and imaged at the confocal.

## Imaging, quantifications and statistical analyses

For whole-mount and cryosectioned samples, imaging was performed using a Leica SP5 confocal microscope with a resolution of 1024×1024 pixels, utilizing either a 20x or 40x objective with a Z-step size of 2 μm. Images were analyzed using Fiji with the Cell Counter plugin. Plot and statistical analyses were done using R, with ggplot2 and ggsignif packages. For experiments that were analyzed in wholemount, each dot represents an embryo. For cryosections, each dot represents a section, and the total number of embryos is indicated in the figure legend. For the *Cxcr4* experiment, each dot represents a cell, and the total number of embryos is indicated in the figure legend. For each condition, a minimum of 4 embryos were analyzed and for each embryo or section, a minimum of 250 cells were counted, leading to a total number of 1000 to 10,000 cells per condition. Experiments with two different conditions were compared using Wilcoxon-Mann-Whitney test and experiment with three different conditions with Kruskal-Wallis test associated with Dunn-Bonferroni post-hoc test. NS represents a $p$-value > 0.05, * a $p$-value < 0.01, *** a $p$-value < 0.001, and **** a $p$-value < 0.0001.

## Purification of electroporated single cell

E4.5 electroporated chicken embryos with a ubiquitous dTomato and the 16TF-VNP reporter were screen under a fluorescent binocular, and electroporated limb buds were quickly dissected and incubated with 500 μl of pre-warmed Dispase (1.5 mg/ml in DMEM/10 mM Hepes),

pipette up and down 10 times and incubated 15 min at 37 °C. The sample was homogenized every 5 min then 500 µl of pre-warmed Trypsin (0.05% in DMEM) was added to the tube, homogenized and incubated 3 min at 37 °C. Samples were then transferred into a 15 ml falcon tube, and the reaction was stop with 10 ml of Hanks buffer (for 100 ml: 10 ml of HBSS 10X, 250 mg of BSA, 1 ml of Hepes 1 M, in sterile ddH₂O), homogenized and centrifuged 10 min at 500 g. The pellet was re-suspended in 4 ml of Hanks buffer and filtered with a pre-humidified 40 µm sterile filter and re-centrifuged 10 min at 500 g. The final pellet was re-suspended into 250 µl of Hanks buffer and added to 250 µl of Hanks buffer in a pre-humidified FACS tube. For sorting, we added DAPI (1/1000) in the final Hanks buffer solution and prepare a sample containing non-electroporated tissues and non-electroporated tissues stained with DAPI to calibrate the sorting. Cells were then sorting according to the dTomato fluorescence and collected into Hank's buffer. An example of the fluorescence-activated cell sorting (FACS) gating strategy is provided in Supplementary Fig. 12. For the single-cell RNA-seq experiment, a total of 6 electroporated limb buds were pooled together in the same tube.

## Single cell RNA-seq analyses of chicken embryo

To profile cell-type composition after electroporation, we performed single cell RNA-sequencing (scRNAseq) on chicken embryo limb bud electroporated with the CAGGS:dTomatoNLS and the 16TF:VNP plasmid (see section above). Data demultiplexed in fastq format was checked with fastqc and fastq_screen to control quality sequencing. Sequencing reads were processed with STAR[116] (v.2.7.9a) to align and quantify data. Genome and transcriptome used is GRCg6a.105 from Ensembl database & custom references gtf and fasta files corresponding to the plasmids used. We merged both datasets in one. The downstream treatment for analysing separately and integratively the data used Seurat package (v5.1.0)[117] after filtering was *SCTransform* with *glmGamPoi* method[118] and a regression of percentage of mitochrondrial, PCA on SCT assay. We determined Doublet barcode with *scDblFinder*[119] R packages (v1.16.0) on cluster-based approach and *dbr = 0.1*. We selected any features where mean (count $_{sample})_{feature}$ > 0.1. (s)KNN graph build on pca reduction matrix with a dimension depending of sample (we selected the last elbow point where cumulative sums of percentage Standard Deviation > 90% and the n + 1 dimension not exceed <1% supplementary), we used Louvain Clustering with *igraph* method. We used also FFT-accelerated Interpolation-based t-SNE[120,121] (FItSNE) with these parameters: a modification of initialization (first and second PCA dimensions are divided by standard deviation of first PCA dimension multiplied by 0.0001), *perplexity_list* used is a list starting 30 to nCells$_{integrated\ dataset}$/100) and *learning rate = nCells$_{integrated\ dataset}$ / 12*, all PCA dimensions selected from the last step and 1000 iterations to run. Biomarkers are identified after *PrepSCTFindMarkers* on SCT assay and performed with *FindAllMarkers* with MAST test[122]. We filtered filtered biomarkers to conserve only features |pct.1$_{feature/cluster}$ – pct.2$_{feature/cluster}$| > 0.2 and p-adj <0.05. We selected a top5 on each cluster found and sign of log2FC. In total, 2108 cells were sequenced and used for further analysis. We identified the presence of absence of the dTomato and the mVenus transcripts and identified 432 cells where dTomato only was expressed ≥ 15 (~19%) & 55 cells where mVenus was expressed ≥ 15 both (~2.6%). Each cell covered a value range of nCount RNA between 805 to 20000, nFeature RNA between 601 to 4462 and %mitochondrial ≤ 0.7%. Muscle progenitors dataset was isolated after identification of clusters based on genes expression and the same protocol of downstream refinement is used (SCT + PCA + dynamic selection of dimension PCA's). P-values for differential expression between clusters were calculated using the MAST method in Seurat's FindAllMarkers package, which models single-cell expression and accounts for detection rate; values were adjusted using the Benjamini–Hochberg correction. Differential expression between 16TF-VNP⁺ and 16TF-VNP⁻ cells was assessed using

the MAST method in Seurat, which fits a hurdle model combining logistic regression for detection and linear modeling for expression levels. These p-values reflect the significance of expression differences while accounting for cellular detection rate. The scRNAseq data presented in this manuscript have been deposited in NCBI's Gene Expression Omnibus under the accession number GSE283478.

## Analysis of *DN Lef1* phenotypes

Images of E4.5 control and DN Lef1 limb bud were acquires in wholemount on the confocal as described above. 3D acquisitions were transformed into 2D images with a maximal projection in Fiji. For the percentage of dispersion, an oblong shape was drawn around the muscle mass to measure its total area and then a threshold was set to measure the area of all the nuclei using the "Analyze particles" function. The percentage of dispersion was calculated as follow:

$$Percentage\ of\ Dispersion = \frac{(Total\ Area - Sum\ of\ Particles\ Area)}{Total\ Area} \times 100$$

For the center of mass, a line was traced in the middle of the muscle mass, along the proximo-distal axis of the limb and the center of mass was measured using the "Measure" function. The center of mass is defined as the brightness-weighted average of the x and y coordinates of all pixels in the image or selection. Therefore, if the repartition of the fluorescence is homogeneous along the proximo-distal axis the center of mass should be around the middle of the region of interest. The center of mass was normalized by the total length of the muscle mass. A value around 50 represents a homogeneous repartition.

$$Ratio = \frac{YM\ of\ Center\ of\ Mass}{Total\ Length\ of\ the\ Muscle\ Mass} \times 100$$

## In ovo cell grafting

7 million HEK-293 cells (#CRL-1573, ATCC) were plated in 10 cm culture-plates with DMEM / 1% Penicillin-Streptomycin / 10% FBS. The following day, 7.5 µg of the CAGGS:mbdTomato plasmid in combination or not with 7.5 µg of the CMV:Wnt1-Myc plasmid were mixed with 1.5 ml of Optimem and 150 µl of Polyethylenimine (PEI, #408727, Sigma-Aldrich) and incubated 30 min at room temperature. The plasmid mix was then added dropwise on antibiotic-free HEK-293. A total of two plates for each condition were transfected. The next day, cells were detached by the addition of 3 ml of 0.25% trypsin in DMEM. The reaction was blocked with DMEM / 1% Penicillin-Streptomycin/10% FBS and cells were harvested in a 15 ml Falcon tube, centrifuged, washed with 1 ml of DMEM/1% Penicillin-Streptomycin/10% FBS and centrifuged in 1.5 ml Eppendorf tubes. Supernatant was removed to obtain a thick preparation of cells. Cells were then loaded into a glass capillary (30-0047, Harvard Apparatus) with a Hamilton Microliter™ syringe #805 and injected using a PicoSpritzer injector into E3.5 chicken embryo limb buds. Wholemount embryos were immunostained and cleared as described above before being imaged with a fluorescent binocular. For the detection of the MYC-tag, embryos were cryosectioned, immunostained, and the MYC-tag was revealed using an anti MYC primary antibody (ab32, Abcam, IgG1, mouse, 1/200), in combination with a Goat anti-Mouse IgG biotinylated (1/200, 31800, ThermoFischer) and a Streptavidin coupled with Alexa 488 (S32354, ThermoFischer). The solidity and the circularity of the muscle mass were evaluated using the "Measure" function in Fiji.

## Analysis of the human embryonic limb atlas

Processed data of myogenesis cells from human embryonic limb atlas was downloaded (https://limb-dev.cellgeni.sanger.ac.uk/) and analyzed in python (v3.10.16) with scanpy (v1.10.4), numpy (v2.0.2),

matplotlib (v3.10.0) and igraph (v0.11.8) libraries. Development trajectories were reproduced following the pipeline describe by the authors[90]. Briefly, we performed nonlinear dimensionality reduction using diffusion maps (scanpy tl.diffmap with 15 components), and the neighborhood graph (scanpy pp.neighbors) was realized based on the 15 components of diffusion maps. We used PAGA[123] with force-directed graph from scanpy library to reproduce the abstracted graph of partitions and visualize the development trajectories. From the differential analysis between 16TF-VPN⁻ and 16TF-VPN⁺ cells we selected genes with a $p$-value < 0.05, a Log Fold Change > 2 and pct.2 < 0.6 to filter up regulated genes in 16TF-VNP⁺ cells and not expressed in more than 60% in 16TF-VNP⁻ cells. The Wnt-TCF/LEF signature was composed of genes regulated by TCF/LEF-dependent Wnt signaling in adult muscle stem cells, previously described (Rudolf et al.,[92] Parisi et al.,[91]). We then visualized target genes expression in human sample and were able to score each cell in function of the expression level of each signature gene sets (scanpy.tl.score_genes).

### Ethics
All experiments conducted on chicken embryos were done in accordance with French and European legislation, which does not require specific ethical approval for work on avian embryos (Article R214-88, Code rural et de la pêche maritime; Directive 2010/63/EU).

### Reporting summary
Further information on research design is available in the Nature Portfolio Reporting Summary linked to this article.

## Data availability
scRNA-sequencing data sets for chicken embryos are available at the Gene Expression Omnibys (GEO) under the accession number GSE283478. Source data are provided with this paper.

## Code availability
Code is available upon request.

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

## Acknowledgements
We thank Drs Frank Stockdale, Frédéric Relaix, and Jean-Louis Bessereau for critical reading of the manuscript and insightful discussions on this study. We thank the Centre d'Imagerie Quantitative Lyon-Est (CIQLE) for imaging support. We thank Cyril Dégletagne from the Plateforme de Génomique des Cancers for support with single-cell RNA-seq. This research has been supported by a grant from the Agence Nationale de la Recherche (ANR) and a grant from AFM-Telethon (MyoNeurALP). GT has received a PhD scholarship from the Ecole Normale Supérieure de Lyon (CDSN).

## Author contributions
Conceptualization: G.T., C.M., F.L.G. Methodology: G.T., C.M. Formal analysis: W.J., F.L.G., S.J. Investigation: G.T., V.M. Funding acquisition: C.M., F.L.G. Supervision: C.M., F.L.G. Validation: G.T. Writing—original draft: G.T., C.M. Writing—review and editing: C.M., G.T.

## Competing interests
The authors declare no competing interests
