## [Transparent Peer Review file · Nature Communications]

Early lineage segregation of primary myotubes from secondary myotubes and adult muscle stem cells.

Corresponding Author: Professor Christophe Marcelle

Version 0:

Reviewer comments:

Reviewer #1

(Remarks to the Author)

In this paper, Toulouse and colleagues use sophisticated in vivo tracing systems in the chicken embryo to demonstrate the existence of two populations of myogenic progenitors in the developing limb. They show that one population activates Wnt signaling and corresponds to the precursors of primary myofibers, while the Wnt-negative population gives rise to secondary myofibers and satellite cells. They describe a thorough characterization of these two progenitor populations and their descendants during myogenesis. Finally, they perform single cell RNAseq analysis of fluorescent limb myogenic progenitors (including the two populations) whose precursors were electroporated with reporters while in the epithelial dermomyotome. This shows little difference between the two populations, but allows them to identify CXCR4 as a gene specifically upregulated in the primary myogenic progenitors. They then speculate that it may play a role in the differential migration of these precursors.

Overall, this is a remarkable study that addresses the long-standing question of primary and secondary myogenesis. This work identifies, for the first time, a selective marker that allows the two populations of progenitors to be clearly distinguished, demonstrating that they coexist in the limb and yet give rise to different sets of derivatives. Remarkably, these two populations appear to show little differences at the transcriptome level. The quality of the work and illustration is remarkable, and in my opinion this is a landmark paper that belongs more to Nature than to Nature Comms. I strongly support publication, provided that the comments below are adequately addressed.

The weakest part of the paper, in my opinion, is the last part dealing with the role of CXCR4.

Since the two populations of progenitors are mixed, the role of CXCR4 in promoting differential migration of the primary progenitors is unclear. Furthermore, this is not addressed at the functional level.

Could the authors provide more convincing differences between the two populations based on the single-cell RNAseq analysis? I found the scRNA analysis of the two progenitor populations very interesting but a bit superficial. It might be worth trying to explore these data further and better describe the trajectories of the cell populations. Also, relating this to mouse or human data (there is a very complete dataset of human limb development) would be very interesting and a valuable addition to the paper.

It would also have been very interesting to know if these two populations are also observed in epaxial muscles. Given the tools and skills developed by this group, it seems like a limited endeavor and this would generalize their findings to primary and secondary myogenesis.

Reviewer #2

(Remarks to the Author)

Reviewer comments:

The manuscript, titled Early lineage segregation of primary myotubes from secondary myotubes and adult muscle stem cell, provides intriguing insight into the origin of primary and secondary myofibers using elegant in vivo studies in the chicken embryo. The authors find that two populations of myogenic progenitor cells, marked by the expression or absence of TCF-LEF/ β -catenin signalling reporters, form primary and secondary myofibers respectively. Interestingly, both populations are present from the beginning of progenitor cell migration from the VLL and lineage tracing experiments find that only 13% of PAX7+ cells express the TCF-LEF/ β -catenin reporter by E16.5. Whether this population is maintained in adulthood remains

an interesting question. The authors also propose that canonical Wnt signalling is a positive regulator of Cxcr4 expression. While the data is convincing, the manuscript requires significant editing to improve readability and transparency. Additional experiments and/or details are required. Detailed points are listed below.

Major point #1: We recommend that the manuscript be carefully edited to provide clearer background information and experimental detail. Examples include:

- At the beginning of the results section (lines 98-102), the authors introduce their previous findings that form the foundation for the current body of work. However, context and introduction of the TCF-LEF/ β -catenin construct would improve readability. For instance, a brief descriptor of "...this reporter (named 16TF-VNP)..." and greater detail in reference to "...muscle forming domains." Sufficient information in the results section to understand the data should be provided.
- To make the manuscript more accessible to general scientific audiences, the authors might consider including brief background information on topic such as Wnt signalling relating the TCF-LEF/ β -catenin construct, the canonical Wnt ligands that regulate limb/muscle development, and a brief description of somites specification. For example, the lateral border of somites is termed VLL, but the authors do not define this acronym in the main text, only the figure legend.
- Figure legends require editing. Examples include:
 - o Letters inaccurately refer to figures. For example, lines 565-579, from "(D) is an enlargement of (C)."
 - o Errors: missing punctuation; sentence case not consistent; arrows, stars, and cell outlines are not all defined; colour referencing mistakes (i.e. Figure S8, "MYOG (magenta..." should say cyan; and different time points are used in the figure legend and illustrations.
 - o Considering the many constructs, fluorophores and time points, in addition to different fluorophore localization (nuclear versus cytoplasmic), clearer separation and labelling of immunofluorescence images and proof reading for inconsistencies in reporting will improve clarity. For example, labelling Figure 5J and 5K with the relevant electroporated construct.
 - o The use of bold text for sub-figures is inconsistent and the repeated lettering is confusing.
- Greater experimental detail is needed. Examples include:
 - o While readers can infer that the embryos illustrated in Figure 2 were electroporated at E2.5, please clearly describe the experiment. "Dorsal view of confocal stacks of E7.5 limb buds electroporated with an ubiquitous nuclear dTomato, the 16TF-VNP reporter and stained for PAX7" (Lines 540-541) suggests that the limb buds themselves were electroporated. Please indicate in results and figure legends. This suggestion applies to multiple points throughout the text.
 - o Significant clarification and editing are required in reference to the DN Lef1 experiments and Figure S8. Lines 352-353 are unclear. In the following: "we repeated the experiment, activating the DN Lef1 expression only after progenitors had migrated out of the VLL and examined embryos at E6.5," the term activation is not defined. Is the text referring to activation of the lineage trace with dox or expression of the dominant negative Lef1 construct? A Tet-on inducible DN Lef1 is described in the S8 figure legend and is labelled in S8J but is not mentioned in the results section or the methods. Also, was the construct electroporated near the VLL to target migrating progenitors or in the limb bud? Please clearly describe the constructs used and the experimental design. Please also add relevant experimental and construct detail to the methods.
 - o The electroporation and analysis timepoints are ambiguous. In Figure S8G-H, the illustration has E4.5 overlayed on the image, while the legend says E6.5. If the timepoint is E6.5, the authors should comment on why the spatial distribution observed in S8L is not observed. The manuscript text should be edited to clarify these experiments.
 - o The manuscript mentions that Wnt-expressing cells were injected into the early limb bud to test whether expression of Wnt controls myogenic cell migration, but the cell lines and Wnt ligand are only shown in supplemental figure and methods.
- Edit text for grammatical errors, repeated words and figure referencing. For example, should line 163, read "Fig. 2A-D"? Present and past tenses are also used interchangeably in results section.

Point #2: DN Lef1 validation

- Please show or reference the validation of Wnt β -catenin-dependent signaling inhibition using the DN Lef1 construct(s) in chicken. Citation #72 was conducted in *Xenopus*. Also, please more clearly describe what constructs were used, and include details on the experiment in the methods section. The use of ubiquitously expressed and inducible constructs is not well explained.

Point #3: Further validation required for the in ovo cell grafting experiment

- The authors indicate that many Wnt ligands are expressed during limb formation. Please describe expression of canonical Wnt ligands in the VLL and the forelimb and explain the use of Wnt3a-expressing cells. What is the endogenous level of Wnt3a in the forelimb? If expressed, does loss of Wnt3a in the limb bud change progenitor cell positioning.
- Please validate that the Wnt3a expressing cells were secreting functional protein in the limb. Is one day sufficient for robust Wnt3a expression?
- Pax7 staining intensity near the cell injection site looks elevated in Figure S9C. Please provide quantitative analysis, such as MFI.

Point #4: More detail required for scRNA-seq experiments.

- In Figure S1C, the authors find that near 100% of electroporated migrating cells are PAX7+. However, the scRNA-seq UMAP presented in Figure 5E shows a large proportion of fibroblasts (darker green) and other unidentified clusters (clusters 4,7,8). Considering the number of cells, these clusters are likely not due to sort contamination.
 - o Considering previous reports that Pax7 lineage cells give rise to fetal FAPs and other cell types in mouse development (Fung et al, Cell Discov, 2022, doi: <https://doi.org/10.1038/s41421-022-00407-0>), can the authors comment on whether the fibroblast cells originate from either the 16TF-VNP positive or negative lineage? Similarly, what is the identity of clusters 4, 7 and 8 cluster in Figure 5E? Do these cells arise from a specific subpopulation of PAX7+ progenitors?
 - o Please more clearly describe the processing steps in the results, and explain which populations were excluded. The non-myogenic cells likely arise from PAX7+ progenitors, thus understanding which lineage they arise from is of interest.

- Please clarify in the text that the 10X experiment was conducted of limb progenitors labelled at E2.5, as the statement “we adopted a transcriptomic approach, performed on limb progenitors co-electroporated with a ubiquitously-expressed nuclear dTomato and the 16TF-VNP reporter...” suggests the limbs themselves were electroporated. See commons from Point #1. This would also impact the interpretation of the non-myogenic electroporated cells that were observed by scRNA-seq.
- Please specify what single cell RNA-seq platform/sequencer was used in the methods. Please more clearly explain what analysis software was used (including versions) and include citations for packages. Please also provides tables with cluster markers and differentially expressed genes.
- Was single nuclear or single cell RNA-sequencing performed? In reference to lines 140-141 of supplemental information file.
- Please provide examples of sort strategy.

Point #5 Information and data required for the Cxcr4 HCR FISH experiment.

- In reference to Figure 5J-L and lines 404-414, the experimental design is unclear and is missing from the materials and methods section. The Cxcr4 HCR FISH experimental protocol should be added to the materials and methods section. How were the probes designed/validated?
- If progenitors remain in a proximal position to the VLL at E4.5 following DN Lef1 electroporation at E2.5, the authors should indicate where the cells analysed were spatially located. The manuscript indicated myoblasts were analysed - was this done on cross sections, whole mount or isolated cells? Was the control VLL of brachial somites electroporated with the CAGGS:dTdtNLS construct? If so, why is tdT expression missing?
- The authors should include nuclear staining and provide additional images of the analysed cells.
- As indicated earlier, the number of cells analysed should also be indicated.
- CXCR4 should be changed to Cxcr4 in the results section.

Minor comments:

- Please include nuclear staining in main figure staining panels. Please also include stains that highlight limb and somite structures in the main figure, such as what is shown in Figure S1D.
- All data is presented as proportions. Please either provide raw quantifications or indicated how many cells were assessed. For instance, in Figure 5J-L.
- Please include statistical tests in figure legends. Also, please provide justification or assess parametric assumptions to explain why all analysis was done using non-parametric tests. In some analysis, the authors test more than two variables, and the Kruskal-Wallis does not test for interactions between two factors.
- Please report group sizes in figure legends or methods.

Reviewer #3

(Remarks to the Author)

The manuscript by Toulouse et al tries to delineate the origins of the myotubes that are generated during the primary and secondary myogenesis waves during avian embryonic development. They identify the presence of TCF/LEF reporter positive and negative myogenic progenitors, employing several novel TCF/LEF reporter constructs, which according to them contribute to the distinct myogenic lineages during avian development. They generate the TCF-Trace Tet-on based construct to track avian embryonic myoblast fate, in some elegant experiments. Finally, they carry out a single cell RNA-sequencing experiment using TCF/LEF reporter positive and negative cells to identify differences between them. To explain the differences in spatial distribution of these cells, the Authors identify the role of the chemokine receptor CXCR4 known to regulate muscle progenitor migration.

The manuscript addresses a longstanding and important question in muscle biology, namely the differences in the origin of myotubes derived during the two phases of embryonic myogenesis - primary and secondary myogenesis. The Authors distinguish between the two waves of developmental myogenesis by implicating the Wnt pathway as a distinguishing signal, specifically using TCF/LEF reporter assays. Further, the TCF-Trace Tet-on system that is devised and described in this manuscript is a novel tool that is of value. However, the mechanistic part of the manuscript, where the Authors claim a role for the CXCR4 signaling in regulating the spatial distribution of myogenic progenitors is weak and needs strengthening. Overall, the manuscript may be revised to address the specific comments below and reconsidered for publication thereafter.

Major comments

1. What are the cells with the green signal (mVenus) lying outside the area marked as somites in the more distal side of the proximo-distal axis in Figure 1C-E? Can an overview picture (low magnification), maybe stitched image of several fields, showing multiple somites be provided for Figure 1F-I, as was provided for Figure 1C-E? This would clearly demonstrate the point the Authors are trying to make.
2. Is there any particular reason why dTomato is used instead of TagBFP in Figure 1I? What is the VLL timepoint and how is the percentage of 16TF-VNP+ cells calculated in Figure 1K? Why is the percentage of 16TF-VNP+ cells very low at E3 (Figure 1K), when it looks quite high in Figure 1C-D? It would be good to include the 16TF-VNP+ cells at E7.5 in the graph 1K.
3. The Myf5 staining in Figure 2A-B shows some cells with a stronger Myf5 signal and others with a weaker signal (can be seen clearly in the last panel of B); why is that and is there any preferential mVenus signal in the nuclei with strong vs weak Myf5 signal?
4. The MyoD labeling in Figure 2C-D is not very convincing, with a lot of variability in the size of the signal between nuclei. It is not clear whether some of the MyoD signal is background. Comparing panels 2B to 2D, why is there a lot more of the mVenus+ cells in D as opposed to B, although both panels are from the same stage?
5. In Figure 2, while all mVenus+ cells are Myf5+ and Pax7+, there are a lot more Myf5+ and Pax7+ cells that are not

mVenus+; why is that? Are these cells of a different origin?

6. In Figure 3A, how is Cre under the control of the rtTA element? Is the Cre under a TetO or similar tetracycline-responsive regulatory element? This needs to be clearly depicted for clarity.

7. The Authors mention that the dTomato+ neural cell numbers are comparable between the non-destabilized rtTA construct (Figure 3E) and the destabilized/translation enhanced rtTA-PEST IVS/Syn21/p10 construct (Figure 3I); however, this does not seem to be the case in the representative images provided, where the number of dTomato+ cells seem higher in the non-destabilized rtTA construct (Figure 3E). A quantification of the proportion of dTomato+ cells in both instances would be useful here.

8. The purpose of the experiment shown in Figure 3J-M is unclear; since the 16xTCF/LEF binding site construct is giving a robust response with the non-destabilized rtTA construct to induce dTomato expression, it is obvious that a ubiquitous CAGGS promoter would do the same; also, why is this experiment analyzed at a later stage (E4.5) than what is shown in Figure 3D-I, which makes it difficult to compare? A better control would have been to compare the rtTA construct (Figure 3E) to the rtTA-PEST IVS/Syn21/p10 construct (Figure 3I) with respect to dTomato expression at a later stage such as E4.5, which would be a logical progression towards the data shown in Figure 4.

9. In Figure 4, panel B is not clearly explained in the figure legend. The panel labeling is incorrect in Figure 4; I think panel B is referred to as C and all panels from hereon are incorrectly labeled, based on the figure legend.

10. The high proportion of TCF-Trace negative progenitors at E7.5 in Figure 4B-D could be due to the inefficiency in labeling as seen with the rtTA-PEST IVS/Syn21/p10 construct compared to the rtTA construct (Figure 3E, I); as discussed in the point 8 above, a comparison of the rtTA construct (Figure 3E) to the rtTA-PEST IVS/Syn21/p10 construct (Figure 3I) with respect to dTomato expression at later stages is required to resolve this.

11. In Figure 2G-J, almost all 16TF-VNP+ cells are PAX7+ at E6.5; therefore, how do the Authors explain the high proportion of PAX7+ cells being TCF-Trace negative at E16.5, as shown in Figure 4K-L?

12. In Figure 5J, why is there no dTomato expression as compared to 5K? The data provided on CXCR4 is preliminary and is not sufficient to conclude that it is the only pathway regulating the spatial distribution of primary progenitors.

13. The replicate numbers and statistical methods used should be clearly described.

Version 1:

Reviewer comments:

Reviewer #1

(Remarks to the Author)

My comments have been appropriately addressed in this revised manuscript. The comparison of the chicken scRNAseq data to the human dataset is particularly welcome and significantly broadens the conclusions of the manuscript. Overall, I find this an important piece of work shedding light on a poorly understood aspect of skeletal muscle development and strongly support publication as is.

Reviewer #2

(Remarks to the Author)

The manuscript is now acceptable for publication!

Reviewer #3

(Remarks to the Author)

The Authors have addressed the comments wherever needed and provided detailed explanations in the rebuttal. In my opinion, this version of the manuscript may be accepted for publication.

Point-by-point responses to reviewers

We would like to thank all reviewers for the time they took to review our manuscript and for their insightful comments.

We are extremely pleased and thank all have found much interest in our manuscript, in particular reviewer 1 is "strongly recommending the publication" of our paper, which he finds is "a remarkable study", and a "landmark paper", "fitted for Nature". Reviewers 2 and 3 found that our in vivo approach (using the TCF-Trace tool we designed) was "elegant" while R#2 found the data "convincing" (reviewer 2) and R#3 mentioned that it "addresses a longstanding and important question in muscle biology".

Here below is a summary of the major changes, including new experiments, that were made to the manuscript, followed by a paragraph describing the additional necessary changes and finally a point-by-point response to all reviewers' comments. We have renumbered the comments in the rebuttal for more clarity.

Major changes

New experiments, new figures:

- i) To address the **comment 2) of reviewer #1 and comment 16) of reviewer #3, that found that our transcriptomics analysis should be more detailed, we have performed** (as suggested by reviewer #1) **a comparative analysis between a single-cell RNA-seq dataset of human embryonic limb development** (Zhang et al. *Nature* 635: 668-, 2023) and our results. Among the tissues constituting the human limb, Zhang and colleagues have identified, within the myogenic population, two co-existing lineages with distinct developmental trajectories. Since they did scRNA-seq and not snRNA-seq, they could not analyze myofiber nuclei, and therefore that have not connected these two populations to primary and secondary myogenesis. Based on the available literature, these authors have named them "embryonic" and "fetal" myogenesis. Despite notable differences between the limb developmental stages in chicken and in human (the latter being significantly more advanced) the comparison of their data with ours uncover striking similarities, **with a number of genes specific of TCF/LEF-responding cells in chicken (i.e. in primary myogenic progenitors) that are also over-expressed during "embryonic" myogenesis in human, suggesting that they represent the same population of cells.** Important is the **identification of CXCR4 over-expression in this population also in human embryo**, supporting the analyses we have done in our manuscript. The results of this analysis are presented in a Supplementary Figure 13. We also added a Supplementary table (Supplementary table 2, line 411-412) with a complete list of genes that were up and down-regulated in the TCF/LEF-responding cells. **This analysis considerably reinforces our conclusion that two distinct myogenic lineages with different transcriptional signatures co-exist during limb myogenesis.** Our lineage tracing data completes this observation, identifying the developmental fate of the two populations as **primary myotubes for the first and secondary myotubes with adult muscle stem cells for the second.**
- ii) We have addressed the **comments 10) of reviewer 2, that asked us to validate the WNT-expressing cells** we have used to address WNT function during migration of progenitors of the primary myogenic lineage. After many failed attempts to detect WNT3a protein by IHC with WNT3a-specific commercial antibodies, we resorted to a new approach. It consisted in **using cells transfected with a Myc-tagged WNT1 construct, and injected into the muscle masses of the developing chicken embryo.** The functionality of the Myc-tagged WNT1 construct we

used has been validated by others and we showed that the cells that were injected in the limbs expressed WNT1, using a Myc-specific antibody. The result is essentially the same as the one we had shown before (no trophic function of WNT during migration), but the molecular tool we used is now validated. We hope this reviewer will find this convincing. The results of this experiment are shown in Supplementary Figure 10.

iii) We have also added a Figure (Supplementary Figure 12d), which illustrates the comparative expression of all 35 genes over-expressed in the 16TF-VNP+ population relative to the 16TF-VNP- population (not provided in the previous version of the manuscript).

Additional changes:

- To reflect the impact of the transcriptomics analysis we performed on human embryonic limbs (see above), we have placed it in a separate chapter, entitled: "*A common developmental program underlies chicken and human limb primary myogenesis.*" (line 385).

- To acknowledge the work that was done to perform this transcriptomics analysis, we have added a new author (Dr Sabrina Jagot), as 2nd co-author with Mr William Jarassier.

Point-by-point responses to reviewers

Reviewer 1

1) R#1, 1: "*Since the two populations of progenitors are mixed, the role of CXCR4 in promoting differential migration of the primary progenitors is unclear.*"

Response: This reviewer rightfully points out that at early stage (e.g. E4.5) the population of progenitors is mixed, with approximately 50% that express the reporter and 50% that do not express it. Therefore, the phenotype that we describe after DN-LEF1-mediated inhibition of Wnt is the composite of cells that were affected by DN-LEF1 and cells that were not. However, this situation is not fundamentally different from that described in the paper from C. Birchmeier in 2005, where *Cxcr4* function was investigated using mouse genetics. In that study, *Cxcr4* loss of function led to defects in limb progenitor distribution (Figure 6 of that paper), particularly in the distal limb muscle masses. Importantly, as we did in our study, not all limb progenitors expressed *Cxcr4* (Figure 3), suggesting the observed phenotype also resulted from a combination of affected and unaffected cells. To address the comment, we attempted to selectively block TCF/LEF activation in TCF/LEF-responding cells by generating a conditional DN-LEF1 construct under the control of a TCF/LEF response element. A GFP reporter linked to DN-LEF1 by a P2A cleavage site served as an internal control for DN-LEF1 activation. However, after three independent electroporation attempts, we did not detect GFP expression, likely due to rapid negative feedback from DN-LEF1, shutting down the TCF/LEF response element.

2) R#1, 2: "*Could the authors provide more convincing differences between the two populations based on the single-cell RNAseq analysis?*"

Response: This reviewer points that there is little transcriptional difference between the TCF-LEF/ β -catenin positive and negative progenitors at E4.5. We were also surprised by this finding. This might reflect i) that at the time of analysis, their separation from a common progenitor in the lateral part of somites (the VLL) is only 24 hours old, or that ii) their fates are quite similar (i.e. both will give rise to muscles). Importantly, the comparison with data obtained from human embryonic limbs supports our findings (see below). However, in addition to the genes related to migration that were shown in Fig5, we now provide a complete list of genes that were up and down-regulated in the 16TF-VNP population (Supplementary Table 2 and Supplementary Figure 12d, detailed in lines 410-417 of the Results section).

The same reviewer suggests that "*exploring a dataset of human limb development would be very interesting and a valuable addition to the paper*".

Response: Indeed, a study on human embryonic limb development using single-cell RNA-seq (Zhang et al. *Nature* 635: 668-, 2023) identified two progenitor cell populations with distinct transcriptional signatures, which could correspond to the primary and the secondary lineages we identified here. As mentioned above, since Zhang's analysis was done by scRNseq and not snRNA-seq, they have missed all infos related to myogenic fibers (that cannot be encapsulated for sequencing), and therefore they have not analyzed myogenic terminal differentiation genes, such as muscle myosins. Therefore, they have not connected these two populations to primary and secondary myogenesis and referred to them as "embryonic" first and "fetal" second myogenesis, also identifying a "transitory state" between the two. We have compared our chicken data to their human data set. Notably, the human limb developmental stages that were analyzed (5-9 weeks of development) are more advanced than the stage (E4) we analyzed in the chicken. Despite this, we found a significant overlap in gene expression: many molecules overexpressed in the chicken primary lineage (Fold Change >2) were also overexpressed during the "embryonic" myogenesis in humans. Among the most prominent were: the Cell Adhesion Molecule 1 (CADM1); The Nectin Cell Adhesion Molecule 3 (NECTIN3), the C-X-C Motif Chemokine Receptor 4 (CXCR4) and the Lymphoid Enhancer Binding Factor 1 (LEF1). Furthermore, genes overexpressed in the chicken primary lineage plotted on the UMAP of human limb data, mainly aligning with the first myogenesis and the transitory states identified by Zhang and colleagues. This analysis reinforces our conclusion that two distinct myogenic lineages with different transcriptional signatures co-exist during limb myogenesis in amniotes. The identification of CXCR4 further supports the hypothesis that CXCR4 is an important molecular actor in the primary lineage in chicken and human limb myogenesis.

Finally, we mapped genes involved in Wnt signaling during myogenesis identified in Parisi et al. 2015 and Rudolf et al. 2016 onto the UMAP of human limbs. Strikingly, the vast majority were localized along the first myogenesis trajectory, further supporting the hypothesis that Wnt signaling preferentially characterizes the primary lineage.

This data has been added to our manuscript as a new Supplementary Fig. 13. The results are mentioned in the Introduction section (lines 89-92) and described in the Result section (lines 410-4356. The Methods section has also been edited (lines 809-823). Since we believe the results of this transcriptomics analysis are far-reaching, we have placed them in a separate chapter, entitled: "*A common developmental program underlies chicken and human limb primary myogenesis.*" (line 385).

3) R#1, 3: He/she mentions that it would be "*very interesting to know if these two populations are also observed in epaxial muscles*".

Response: We believe the two situations are significantly different. We have extensively published on the role of Wnt during early myogenesis in the epaxial domain (Marcelle et al., *Development* 1997; Gros et al., *Nature* 2009; Rios et al., *Nature* 2011; Sieiro et al., *eLife* 2016; Serralbo et al., *Development* 2014). Wnt function in this somitic region is complex, as it is implicated in both the spatial organization of the myotome through Planar Cell Polarity and in myogenesis, which is triggered by an unexpected Notch-Snail- β -catenin pathway distinct from classical Wnt signaling. We do not have any evidence that a similar pathway is involved in the cell fate decisions of VLL cells or limb muscle progenitors.

A second key difference between limb and trunk myogenesis is that in the trunk, myogenesis arises from two spatially distinct origins: (i) a population of cells emerging from the epithelial borders of the dermomyotome gives rise to what we have termed the primary myotome, a process triggered by Notch (Gros et al., *Dev Cell* 2004; Sieiro-Mosti et al., *Development* 2014),

while (ii) a second wave of myogenesis originates from the central dermomyotome, induced by Fgfs secreted from the primary myotome (Delfini et al., Dev. Biol. 2009), ultimately generating the bulk of fetal muscles and satellite cells (Gros et al., Nature 2005). In contrast, in the limb, the primary and secondary lineages share a common origin, the population of progenitors migrating from the VLL.

For all these reasons, we do not believe that our observations in the limb can be directly transposed to the trunk, even though we strongly suspect that myogenesis in both regions follows a similar logic, based on two successive waves (primary and secondary). Demonstrating this, however, will require identifying robust markers that distinguish the first and second lineages throughout the entire body—an ambitious goal that represents one of the main aims of a newly funded grant we have just started in collaboration with Pascal Maire, a colleague in Paris.

Reviewer #2

R#2, 1:

1) He/she suggested that *"context and introduction of the TCF-LEF/ β -catenin construct would improve readability"* and he/she invited *"greater detail in reference to muscle forming domains"*.

Response: We have added details on limb muscle formation from progenitors located in the VLL in the first paragraph of the result section. The term “VLL” is also defined (lines 105-106). A short introduction to the reporter construct is also present in the same paragraph, which we hope provides sufficient context for readers to understand its main characteristics (lines 102-109). Additionally, we now provide schematics representing all steps of forelimb myogenesis in the chicken embryo (Supplementary Figure 1a). In addition to the more complete description available in the Materials and Methods section (lines 560-570), we hope these modifications improve the text’s clarity and readability.

2) He/she suggests that *"the authors might consider including brief background information on topic such as Wnt signalling relating the TCF-LEF/ β -catenin construct, the canonical Wnt ligands that regulate limb/muscle development, and a brief description of somites specification"*.

Response: i) A concise explanation of TCF/LEF binding sites and their relevance to Wnt signaling is available in the Materials and Methods section (see above). We have also added a paragraph in the Results section that briefly describes the contribution of somites to limb myogenesis (lines 102-109) and a new supplementary figure describing chicken forelimb myogenesis (see point 1)

ii) The topic of Wnt signaling in limb muscle development is poorly known. Wnt expression pattern in limb development is very complex. A comprehensive ISH survey in mice (Witte et al., *Gene Expression Patterns*, 2009) identified 15 of the 19 Wnt genes and 10 of the 13 Wnt antagonists expressed during limb outgrowth. They have, however, not looked at the expression data of neither the 10 Fzd receptors, nor the many co-receptors (Lrps, Ror, Ryk...). While a similar survey does not exist in birds, published work (e.g. Geetha-Loganathan et al., *Organogenesis* 2008; Kawakami et al. *Cell* 2001; Kengaku et al., *Science* 1998; Church & Francis-West, *Int J Dev Biol* 2002,...) and data from the GEISHA avian in situ hybridization database suggests comparable complexity. For such a variety of Wnts to trigger specific answers, these molecules likely act very locally. Wnts have been shown to be highly hydrophobic through post-translational modifications (Willert et al., *Nature* 2003; Gross and Boutros, *Curr. Op. Gen. Dev.* 2013), a property that does not allow them to act at a distance,

unless active transport is taking place (through exosomes, High Density Lipoproteins, or cytonemes) or active cell migration as we have shown years ago (Serralbo et al. Development 2014). Given the large number of interacting Wnt pathway members during limb formation, a systematic conditional knockout approach in mice would be necessary to dissect the function of every Wnt signaling and interacting molecules in limb muscle development—a massive endeavor that remains largely unexplored.

To the best of our knowledge, only four studies have specifically investigated Wnt function in embryonic limb myogenesis (Anakwe et al., Development 2003; ten Berge et al., Development 2008; Abu-Elmagd et al., Dev. Biol. 2010; Hutcheson et al., Genes & Dev. 2009). Anakwe et al. demonstrated that Wnt5a and Wnt11 inversely regulate the proportions of slow vs. fast myosin fibers without affecting overall muscle mass. Ten Berge et al. found that Wnt3a can redirect chondrogenic precursors toward a muscle fate. Abu-Elmagd et al. overexpressed the Wnt-inhibiting molecules DNLEF1 or DKK1 in early chick limb buds and observed no effect on MRF activation. Hutcheson et al. deleted β -catenin in transgenic mice and found no impact on early myogenesis, although they did observe a role in the emergence of slow myofibers at later stages. Importantly, those experiments were done at a time when the presence of the two distinct populations we identified was unsuspected, possibly leading to incomplete or confusing conclusions. Moreover, key methodological differences make direct comparisons with our analyses challenging. These include the use of replication-competent viruses, which equally spread to target and non-target cells, in Anakwe's study or longer experimental timeframes in the studies by Anakwe, ten Berge, and Hutcheson. Despite this, it is notable that the findings of Abu-Elmagd et al. and Hutcheson et al. align with our conclusions, supporting the idea that TCF/LEF signaling is not required for the transcriptional activation of myogenic regulatory factors. We added the findings of Abu-Elmagd et al. in our discussion (lines 541-545)

3) He/she mentions a few mistakes in Figure legends and errors in the text.

“Letters inaccurately refer to figures. For example, lines 565-579, from “(D) is an enlargement of (C).”

Response: This is a mistake, thank you for pointing it out, we updated the figure legend of the Fig4 with the right panel (lines 870-890)

4) *Errors: missing punctuation; sentence case not consistent; arrows, stars, and cell outlines are not all defined*

Response: Thank you for pointing this out, we added description of arrowhead in the legend of Fig4 (lines 879, 883 and 888) and for the asterisk in Fig5B (line 894)

5) *“colour referencing mistakes (i.e. Figure S8, “MYOG (magenta...” should say cyan; and different time points are used in the figure legend and illustrations.”*

Response: This is a mistake, thank you for pointing it out. It was corrected, however FigS8 is now FigS9

6) *« For example, labelling Figure 5J and 5K with the relevant electroporated construct.”*

Response: This is a mistake, thank you for pointing it out, we updated the Fig5J-K with the relevant constructs.

7) *He/she suggests that clarifications on experimental procedures should be added regarding the Figure 2 and electroporation.*

Response: We updated the Fig2 with a schematic that depicts the experimental procedure we followed (line 843). We used the formulation “electroporated limb bud” throughout the

manuscript and figure legends to define a limb bud that contains myoblasts electroporated with our constructs. Electroporation targets only epithelial cells. Therefore, muscle progenitors can only be electroporated within the epithelial ventro-lateral lip of somites. We understand that this formulation might be confusing, so we added a supplementary figure with a large field picture of a limb bud containing electroporated myoblasts to better visualize the muscle mass inside the limb bud (Supp Fig1b,c)

8) R#2, 2 & 3: He/she would like a *validation of the DN LEF1 construct* we have used and more infos on how we triggered its activity.

Response: i) The Dominant Negative form of LEF1 that was used in our study is derived from a chicken LEF1 cDNA (described in Kengaku et al., 1998, and obtained from Cliff Tabin's lab). It contains a mutated form of the chicken LEF1 cDNA that binds to DNA but not to β -catenin (similar to the Molenaar et al. 1996 design that was tested in *Xenopus*), thereby acting as a dominant-negative mutant. Moreover, we have shown in Barzilai-Tutsch et al. (eLife, 2022, Supp Figure 1 K-O) that this construct very effectively inhibits the response of the Wnt reporter we have used. We have clarified these details in the Material and Methods and in the Reference sections (lines 617-625).

ii) *Related to the point above, this reviewer would like to have clarifications on the text in Lines 352-353, describing the mode of activation of the inducible DN LEF1.*

Response: Due to the constraints on the manuscript's length, the constructs that we used might not have been sufficiently described. The DN LEF1 cDNA (see above) was cloned into two plasmids. In the first plasmid, DN LEF1 is driven by a ubiquitous promoter (CAGGS). It is therefore expressed from the very start of the experiment, when progenitor cells are still located in the VLL. This has given us the result we have shown in Figure 5 a,b. As our analysis of TCF/ β -catenin-dependent signaling had demonstrated that VLL cells are actively responding to this pathway, we were concerned that the phenotype we obtained (in Figure 5 a,b) could be due to a combination of activities, that in the VLL (which we believe is unrelated to the myogenic lineage), and the one in the limb, the focus of our work. We therefore devised a second inducible plasmid, where DN LEF1 was cloned downstream of a bi-directional Tetracyclin Responsive Element (named pBI), activated by rtTA in the presence of Doxycyclin (i.e. a bona fide Tet-on system). This system also results in the production of a nuclear dTomato concomitantly to the DNLEF1. The term activation therefore refers to the time when Doxycyclin was added, triggering DN LEF1 transcription. The protocol we used in this experiment was to activate DN LEF1 transcription from E4.5 when all muscle progenitor cells from the VLL are all within the limb mesenchyme. This was done by adding doxycyclin at E4.5 and E5.5. Embryos were then examined at E6.5, and the result is shown in Supplementary Figure 9 n,o. The result obtained is coherent with that obtained in Figure 5 a,b. We understand that the procedures that were used and the multiplicity of vectors is somewhat confusing for researchers that are not familiar with those techniques. We have thus revised the Materials and Methods section (line 617-625), as well as the figure legends (Supplementary Fig.9) and main text (line 356-362) to clarify those approaches.

9) He/she mentions that *"the electroporation and analysis timepoints are ambiguous. In Figure S8G-H, the illustration has E4.5 overlaid on the image, while the legend says E6.5. If the timepoint is E6.5, the authors should comment on why the spatial distribution observed in S8L is not observed.*

Response: This is our mistake, thank you for pointing this out; the legend has been modified to "E4.5". The MYF5 staining is more intense in the center of the muscle mass than at the edge,

therefore for full transparency we choose to display an inset representing both the center and the edge of the muscle mass. The Supplementary Fig8 is now the Supplementary Fig9

10) He/she mentions that "*The manuscript states that Wnt-expressing cells were injected into the early limb bud to test whether expression of Wnt controls myogenic cell migration, but the cell lines and Wnt ligand are only shown in supplemental figure and methods*". This remark echoes the point #3 that "*Further validation required for the in ovo cell grafting experiment*" and "*Please validate that the Wnt3a expressing cells were secreting functional protein in the limb.*"

Response: To provide evidence that the Wnt3a-expressing cells (purchased from ATCC), effectively express this molecule, we have unsuccessfully attempted to stain them by IHC with a commercially available Wnt3a-specific antibody. Resorting to a different strategy, we used a mouse, Myc-tagged, Wnt1 cDNA construct (McMahon and Moon, Cell 1989), driven by a CMV promoter, to transfect HEK 293 cells. Wnt1, as Wnt3a, is known to activate Wnt canonical signaling in many cellular contexts (See the Wnt Homepage). This construct was shown by McMahon and Moon to be biologically active. The transfected cells were injected into the growing limbs of E 3.5 chicken embryos and analyzed one day later. Importantly, we have observed that these cells robustly express Wnt1, illustrated by the staining with an anti Myc antibody. As we had observed previously, the Wnt1-expressing cells did not change the general organization of the muscle masses, illustrated by the expression of the muscle progenitor marker Pax7. We have made a new figure with these results (Supplementary Figure 10) that replaces the previous one. A description of the vector used and the HEK transfection procedure is found in the Materials and Methods section (lines 629-633 and 790-795). We hope the reviewers find this technical solution convincing.

11) R#2, 4: *He/she would like to have more details on the different populations found in the scRNA seq data: In Figure SIC, the authors find that near 100% of electroporated migrating cells are PAX7+. However, the scRNA-seq UMAP presented in Figure 5E shows a large proportion of fibroblasts (darker green) and other unidentified clusters (clusters 4,7,8). Considering the number of cells, these clusters are likely not due to sort contamination". Additionally, he/she asks us to comment on recent lineage-tracing studies of PAX7+ progenitors in mice, which identified fibro/adipogenic progenitors within this population (Fung et al., Cell Discov, 2022).*

Response: The presence of non-myogenic clusters is expected, as flow cytometry is an enrichment technique rather than a perfect selection process (e.g. Ibrahim & van den Engh, 2007). The identities of these clusters were determined based on transcriptional signatures:

- Cluster 11: Erythrocytes (*HBB*)
- Cluster 10: Early endothelial cells (*VEGFR2*)
- Cluster 9: Macrophages (*CSF1R*)
- Cluster 3: Fibroblasts (*PDGFRA*)
- Clusters 4, 7, and 8: High mitochondrial gene content, likely reflecting dying cells

Apart from fibroblasts (Cluster 3), all non-myogenic clusters are proportionally small. At E4 in the chick limb bud, myogenic cells represent ~0,5% of total limb cells (<https://doi.org/10.1038/s41467-021-24157-x>; Fig 4B). Since electroporation targets ~50% of somite-derived cells, the myogenic population is predicted to be ~0,25% of total limb cells. As myogenic cells constitute 62% of our sorted population (1314 myogenic cells among 2108 cells captured by the scRNA-seq), we can estimate an enrichment of ~250-fold, which is considerable.

Regarding the concern that non-myogenic clusters might be true somite derivatives: while endothelial cells can originate from somites (Eichmann et al., 1993, 1997; Yvernogeu et al.,

2012), there is no evidence supporting a somitic origin for limb fibroblasts or other identified non-myogenic cells. Thus, we interpret these clusters as contaminants, reflecting the inherent limitations of flow cytometry-based sorting.

12) On PAX7+ Progenitors and Lineage-Tracing in Mice

Response: R#2 references Fung et al. (2022), which showed that muscles and fibro/adipogenic progenitors (FAPs) share a common origin. However, in mammals, the dermomyotome gives rise to muscle, dermis, and brown fat, illustrating a well-documented developmental link between myogenesis and FAPs. This study was conducted on whole embryos, making the presence of these derivatives unsurprising.

In contrast, our analysis focuses on the limb, where all available data indicate that limb fibroblasts arise from lateral plate mesoderm, not from the ventrolateral lip (VLL) of the somite. Thus, the findings of Fung et al. (2022) do not contradict our interpretation that the fibroblasts we identified originate from the lateral plate mesoderm, and not from PAX7+ progenitors from the VLL.

13) On Cell Purity

Response: Cell purity is a critical consideration in transcriptomics. Numerous studies (e.g., Ibrahim & van den Engh, 2007) have underscored the limitations of flow cytometry, emphasizing that it enriches rather than selects a population. Purity can be significantly improved by:

1. Stricter gating, though at the cost of yield.
2. Additional sorting rounds, which further reduce yield and viability while increasing cell manipulation, potentially introducing transcriptional artifacts (Machado et al., 2017; van Velthoven et al., 2017).

Our goal was to compare reporter+ and reporter- populations, not to fully characterize all VLL-derived subpopulations. Performing a complete characterization of VLL-derived subpopulations would require additional technological refinements and extensive controls that are beyond the scope of this study.

14) R#2, 5: Please clarify in the text that the 10X experiment was conducted of limb progenitors labelled at E2.5, as the statement “we adopted a transcriptomic approach, performed on limb progenitors co-electroporated with a ubiquitously-expressed nuclear dTomato and the 16TF-VNP reporter...” suggests the limbs themselves were electroporated.

Response: All experiment made in this study were done the same way, by electroporating the epithelial cells located at the lateral border of somites (named the VLL). Thus the limbs themselves were not electroporated. The experimental procedure is described in the material and methods and in Fig5e. We added a supplementary figure for clearer explanation (Supplementary Fig. 1). We modify this sentence if the main text as suggested (lines 389).

15) • Please specify what single cell RNA-seq platform/sequencer was used in the methods. Please more clearly explain what analysis software was used (including versions) and include citations for packages. Please also provides tables with cluster markers and differentially expressed genes.

Response: We updated the method section with the version of the software that was used and we provide citations for packages (lines 739-767). We also added the name of the platform that performed the scRNA-seq in the Acknowledgments section (line 910). We extensively revised the heatmap provided in Supp Fig11 (now Supplementary Fig12b) for depicting more precise

markers of each cluster. We also provided the table of markers genes for each myogenic cluster (Supplementary Table 1, line 399).

16) • *Was single nuclear or single cell RNA-sequencing performed? In reference to lines 140-141 of supplemental information file.*

Response: Single cell RNA-sequencing was performed. At E4.5 the myogenic myoblasts are just initiating their terminal differentiation process, and polynucleated myofibers are not yet present. A regular single-cell RNA-sequencing experiment was sufficient to capture all cells of interest (i.e. from proliferating myoblasts -cluster 2 and 1 Fig. 5F and Supp Fig12 c- to pre-fusing myocytes -expressing *Tmem8c/Myomaker*, Fig5F and Supp Fig12 c)

17) • *Please provide examples of sort strategy.*

Response: A sorting strategy is represented in Supp Fig11.

18) R#2, 5: *"In reference to Figure 5J-L and lines 404-414, the experimental design is unclear and is missing from the materials and methods section. The *Cxcr4* HCR FISH experimental protocol should be added to the materials and methods section. How were the probes designed/validated?"*

Response: Thank you for pointing out this omission. We have added a paragraph on the HCR FISH experiment in the Material and Methods section (lines 682-701). The probe was designed by the Molecular Instrument Co and it was previously tested and published in André et al., 2024 -<https://doi.org/10.1186/s12915-024-01922-0>. The probe mix is composed of a set of 20 small probes that are designed by Molecular Instrument, based on the sequence of the gene of interest in NCBI (here: NM_204617.2). However, the exact sequences of the probes are not available from the company.

19) *"If progenitors remain in a proximal position to the VLL at E4.5 following DN *Lef1* electroporation at E2.5, the authors should indicate where the cells analysed were spatially located. The manuscript indicated myoblasts were analysed - was this done on cross sections, whole mount or isolated cells? Was the control VLL of brachial somites electroporated with the CAGGS:dTdtNLS construct? If so, why is tdT expression missing?"*

• *The authors should include nuclear staining and provide additional images of the analysed cells."*

Response: As described in the methods section (lines 765-784), E4.5 control and DNLEF1 limb buds were clarified in glycerol and imaged in wholemount with a confocal microscope, using a 20x objective. The cells identified by an asterisk in Fig5B are the cells that remained in a proximal position of the limb bud, following DNLEF1 over-expression. For this experiment, control brachial somites were electroporated with a CAGGS:dTomatoNLS plasmid (Fig5A). A picture of a whole limb bud with electroporated myoblasts has now been provided in Supp Fig1b for a clearer spatial representation. Furthermore, we provide additional images of the analyzed cells. If this reviewer requires it, we can include these images in a supplementary figure.

To evaluate the expression of *CXCR4*, we electroporated brachial somites with the CAGGS:dTomatoNLS alongside with the CAGGS:DN LEF1 plasmid. As electroporation results, at best, in 50% of the total cells electroporated, electroporated (dTomato+) myoblasts co-exist with non-electroporated myoblasts (dTomato-). Fig5J is a representative image of non-electroporated cells, immunostained for PAX7 and hybridized with the *Cxcr4* probe. As PAX7 is a transcription factor specifically expressed in the nuclei of muscle progenitors, we

used this to delineate the non-electroporated muscle progenitors. Fig5K is a representative image of electroporated cells with a plasmid coding for a nuclear dTomato alongside the dominant negative form of LEF1, immunostained for PAX7 and hybridized with the *CXCR4* probe. The same strategy has been used in non-electroporated cells. It is noteworthy that the PAX7 staining and the nuclear located dTomato signals perfectly correlate, confirming the pertinence of using the PAX7 staining as a reliable tool to identify the nuclei of muscle progenitors. As DAPI labels all cells, this tends to result in an overwhelming signal under confocal examination. In addition, DAPI would not differentiate between muscle progenitors, mesenchymal or epidermal cells, making the analyses complicated.

We provide here below additional images that could be added as a supplementary figure if this reviewer finds it necessary:

20) Minor Comments of Reviewer 2

Please include nuclear staining in main figure staining panels. Please also include stains that highlight limb and somite structures in the main figure, such as what is shown in FigureS1D

Response: Most of our images are confocal Z-stack (approximately 80x1µm optical sections) of wholemount samples (Fig1, Fig2, Fig3 and Fig5). Myogenic cells represent in between 0,5% to 2,5% of all cells in a limb bud at our stages of interest. Adding a nuclear staining at these stages would have led in too many nuclei represented on the z-stack. As mentioned above, the early stages of myoblasts migration in the limb bud are easy to image in whole mount (Fig1c-e, Supp Fig2a). However, as the embryo and limbs grow, imaging becomes more and more challenging. This is why we decided to section the forelimb at the level of the brachial level from E4.5 to analyze them.

All data is presented as proportions. Please either provide raw quantifications or indicated how many cells were assessed. For instance, in Figure 5J-L. Please include statistical tests in figure legends. Also, please provide justification or assess parametric assumptions to explain why all analysis was done using non-parametric tests. In some analysis, the authors test more than two variables, and the Kruskal-Wallis does not test for interactions between two factors. Please report group sizes in figure legends or methods.

Response: We described the number of embryos analyzed in each figure legend and we revised the methods section to include how many cells were counted on average (lines 704-715). In this study, non-parametric tests were chosen, due to the assumption that the sample data did not follow a normal distribution and exhibited heteroscedasticity. Parametric tests, such as t-tests or ANOVA, can be applied only when the data follow a normal distribution with a homogeneity of variances, conditions that would not necessarily be met for all experiments. Although non-parametric tests may have lower statistical power compared to their parametric counterparts, they are reliable in the presence of skewed data, outliers, or unequal variances, and they offer the only valid approach when distributional assumptions are in question. For Fig. 4h, we used a Kruskal-Wallis test to assess differences in the presence of TCF-Trace+ cells across three conditions: E9.5, E12.5, and E16.5. The two boxplots for each condition (TCF-Trace- in gray and TCF-Trace+ in purple) represent the same data in complementary ways for easier visualization. In other words, the TCF-Trace- values are simply the difference from 100% of the TCF-Trace+ values.

Reviewer #3

1) *"What are the cells with the green signal (mVenus) lying outside the area marked as somites in the more distal side of the proximo-distal axis in Figure 1C-E?"*

Response: These are the first myogenic progenitors migrating out of somites.

2) *"Can an overview picture (low magnification), maybe stitched image of several fields, showing multiple somites be provided for Figure 1F-I, as was provided for Figure 1C-E? This would clearly demonstrate the point the Authors are trying to make".*

Response: These images are stitched confocal acquisitions of the dorsal muscle masses of electroporated limb buds from E4.5 to E7.5. From E4.5 the limb bud was sectioned at the level of the shoulder and imaged in wholemount after glycerol-mediated clarification. From E4.5, all cells originating in lateral parts of brachial somites have delaminated into the limb bud, therefore no fluorescent cells should remain in the trunk region from E5. We provided a picture of an embryo electroporated and imaged under a dissecting scope for a clearer 3D representation (Supplementary Fig1c)

3) R#3 2. *"Is there any particular reason why dTomato is used instead of TagBFP in Figure 1I?"*

Response: The reason is purely technical: we routinely used the CAGGS:dTomatoNLS plasmid as electroporation marker, because it is bright and nuclear (so that there is no doubt on identifying the electroporated cells). The CAGGS:TagBFP plasmid was sometimes used, because it emits light that can be excited and detected with filters used for DAPI and therefore it allows detection of 3 others fluorophores in our current set-up (e.g. Electroporation marker TagBFP: 405nm, 16TF-VNP: 488nm, Immunostaining X: 555nm, Immunostaining Y: 647nm). The E4.5 / E5.5 / E6.5 samples shown in Fig1F-I were also used for immunostaining of PAX7/MYF5 or PAX7/MYOD for analysis in Fig2, this would have not been possible with the CAGGS:dTomatoNLS plasmid in our current set-up.

4) *What is the VLL timepoint and how is the percentage of 16TF-VNP+ cells calculated in Figure 1K?*

Response: At E3, epithelial cells of the lateral part of somites (electroporated at E2.5) initiate their migration laterally into the limb mesenchyme. A large proportion of them are still located within the VLL. Eventually, all VLL cells migrate out of the VLL. To distinguish between these two populations, we changed their name for “E3 epithelial” and “E3 migrating” (Fig1. K)

5) *Why is the percentage of 16TF-VNP+ cells very low at E3 (Figure 1K), when it looks quite high in Figure 1C-D?*

Response: We observed that the fluorescence (i.e. the activity of the reporter) significantly drops in the cells that just exited from the VLL to initiate their migration. The high number corresponds to cells within the VLL, the lower one to cells migrating. As mentioned above, we do not believe that the activities of the reporter observed in VLL cells and in the myogenic progenitors within the limb bud are related.

6) *It would be good to include the 16TF-VNP+ cells at E7.5 in the graph 1K.*

Response: At E7.5, 16TF-VNP+ cells are extremely rare within the muscle mass and are not homogeneously distributed. Given the overwhelming majority of 16TF-VNP- cells at this stage we thought that quantifying the proportion of positive cells would be misleading and biologically irrelevant. Instead, we focus on earlier stages where the reporter signal is more uniform and easily quantifiable.

7) *R#3 3. The Myf5 staining in Figure 2A-B shows some cells with a stronger Myf5 signal and others with a weaker signal (can be seen clearly in the last panel of B); why is that and is there any preferential mVenus signal in the nuclei with strong vs weak Myf5 signal?*

Response: It is true that some muscle progenitors seem more positive for MYF5 than others. Myogenic progenitors within the limb muscle mass are not all synchronized and two adjacent myoblasts can be at different stages of the myogenic differentiation, which could explain the differences observed with the antibody staining. Supporting this, we did observe in our scRNA-seq analyses, clusters displaying higher expression of MYF5 (cluster 0 and 3, differentiating myoblasts 1 and 2, respectively). We observed that 16TF-VNP- cells displayed a higher expression of MYF5 compared to 16TF-VNP+ cells. However, the p-value was high (pval= 0,21 and p_val_adjust=1, see below), explaining why MYF5 was not a hit of the differentially-expressed gene analyses of the 16TF-VNP- and 16TF-VNP+ cells.

8) R#3 4. *The MyoD labeling in Figure 2C-D is not very convincing, with a lot of variability in the size of the signal between nuclei. It is not clear whether some of the MyoD signal is background. Comparing panels 2B to 2D, why is there a lot more of the mVenus+ cells in D as opposed to B, although both panels are from the same stage?*

Response MYOD expression starts at E4.5 (Fig. Supp. 11C; see also MYOD1 on geisha.arizona.edu, Entry 1, Stage 22) and is primarily localized at the center of the muscle mass. Consequently, MYOD protein is barely detectable at this stage. As a result, immunostaining at E4.5 is very challenging, explaining the speckles that are observed. In contrast, by E6.5, its expression is robust and readily detected (Fig. 2h). The difference between the 2B and the 2D panels is likely due to the inherent variability of the electroporation technique.

9) R#3 5. *In Figure 2, while all mVenus+ cells are Myf5+ and Pax7+, there are a lot more Myf5+ and Pax7+ cells that are not mVenus+; why is that? Are these cells of a different origin?*

Response: Electroporation targets at best 50% of the cells. For simplification, the TagBFP (i.e. electroporation marker) channel was not shown in this panel. The mVenus- / MYF5+ / PAX7+ cells are therefore TagBFP+/16TF-VNP- cells or non-electroporated cells.

10) R#3 6. *In Figure 3A, how is Cre under the control of the rtTA element? Is the Cre under a TetO or similar tetracycline-responsive regulatory element? This needs to be clearly depicted for clarity.*

Response: We used the pBI plasmid which contains an element (named pBI) that comprises two opposite minimal promoters under the control of the same tetracycline-responsive elements. This pBI element allows the simultaneous activation of two genes' transcription upon addition of Doxycyclin. However, for this experiment, we drove the expression of only one gene, the Cre. We changed the name of the promoter to TRE (Tetracycline Responsive Element) for simplification (line 612-613 in the method section)

11) R#3 7. *The Authors mention that the dTomato+ neural cell numbers are comparable between the non-stabilized rtTA construct (Figure 3E) and the destabilized/translation enhanced rtTA-PEST IVS/Syn21/p10 construct (Figure 3I); however, this does not seem to be the case in the representative images provided, where the number of dTomato+ cells seem higher in the non-stabilized rtTA construct (Figure 3E). A quantification of the proportion of dTomato+ cells in both instances would be useful here.*

Response: Thank you for this suggestion, we now added a quantification of the percentage of the dTomato+ cells (Fig3j, figure legend line 864), that shows no statistical differences between

the non-destabilized rtTA construct and the rtTA-PEST IVS/Syn21/p10 construct in the presence of dox.

12) R#3 8. *The purpose of the experiment shown in Figure 3J-M is unclear; since the 16xTCF/LEF binding site construct is giving a robust response with the non-destabilized rtTA construct to induce dTomato expression, it is obvious that a ubiquitous CAGGS promoter would do the same; also, why is this experiment analyzed at a later stage (E4.5) than what is shown in Figure 3D-I, which makes it difficult to compare? A better control would have been to compare the rtTA construct (Figure 3E) to the rtTA-PEST IVS/Syn21/p10 construct (Figure 3I) with respect to dTomato expression at a later stage such as E4.5, which would be a logical progression towards the data shown in Figure 4.*

Response: The experiments we showed in Figure 4 are central to our demonstration, as they suggest that there is a portion of limb muscle progenitor cells that never experienced TCF/LEF signaling. An obvious critique of those experiments would be that we observed non-labelled cells because the TCF-Trace system is not sensitive enough, or because a portion of the progenitors have not taken up all plasmids. The experiments shown in Figure 3 D-I are meant to show that the system is very sensitive. The experiment shown in Figure 3J is meant to address the latter concern. It shows that after electroporation of brachial somites with a rtTA tool driven by the ubiquitous CAGGS promoter, the Tet-On / Cre-Lox based system labels nearly 100% of electroporated cells. This demonstrates that all cells have equally taken up all plasmids (5, if we consider the transposase plasmid). We expected that comparing CAGGS:rtTA with CAGGS:IVS-Syn21-rtTAPEST-p10 would yield similar results. Even if the rtTA protein were less stable in the latter construct, the strong activation of the CAGGS promoter would ensure continuous production of rtTA, maintaining comparable expression levels.

We have explained in the manuscript that using the rtTA construct (Figure 3E) is not appropriate to our question, as it would trace cells present in the VLL that activate TCF/LEF signaling within this structure, which is not the purpose of the experiment. We believe the activation of TCF/LEF signaling in the VLL is related to the maintenance of its epithelial integrity (e.g. Linker et al. Dev. 2005) and is therefore unrelated to its function in the limb muscle progenitors.

13) R#3 9. *"In Figure 4, panel B is not clearly explained in the figure legend. The panel labeling is incorrect in Figure 4; I think panel B is referred to as C and all panels from hereon are incorrectly labeled, based on the figure legend".*

Response: This a mistake, thank you for pointing this out, we changed the figure legend accordingly (lines 877-879)

14) R#3 10. *The high proportion of TCF-Trace negative progenitors at E7.5 in Figure 4B-D could be due to the inefficiency in labeling as seen with the rtTA-PEST IVS/Syn21/p10 construct compared to the rtTA construct (Figure 3E, I); as discussed in the point 8 above, a comparison of the rtTA construct (Figure 3E) to the rtTA-PEST IVS/Syn21/p10 construct (Figure 3I) with respect to dTomato expression at later stages is required to resolve this.*

Response: As explained in point 8 above, an rtTA tool cannot be used to address those questions as it would carry the memory of the activity of TCF/LEF signaling from the VLL, likely unrelated to its function in the limb muscle progenitor population.

However, an important issue underlying comments #10 and #8 of this reviewer relate to the following: could the failure to label a portion of the myogenic progenitor population due to a low sensitivity of the tracing system? We have previously observed that the transcriptional response after Tet-on activation is very strong, comparable to the transcriptional response of

the very efficient CAG promoter, that we and many other use. This is particularly true for the so-called Tet-on 3G system (first generated by Clontech, now distributed by Takara), which is both highly sensitive and displays a very low background (Rios et al. Nature 2011; Serralbo et al., Genesis 2013; Véron et al. Dev. Biol. 2015). In our experimental design, the rtTA is driven by the most effective to-date TCF/LEF response elements, boosted by the translation enhancer cassettes we described. The doxycycline inducer is present in the embryos from E4.5 until the very end of the TCF/LEF activity in progenitors (i.e. E7.5). The Cre/Lox system is extremely sensitive, up to a point we had to use self-inactivating (i.e. floxed) Cre constructs to "tame" its activity (Sieiro-Mosti et al., Development 2016). We show in this manuscript that there is a strong difference between the cells that activate the TCF/LEF reporter and those that do not. Finally, we show (with an experiment that compares a stable and an unstable fluorescent reporter protein) that this activation is sustained over extended periods of time. Together with the demonstration that all plasmids enter electroporated cells (see above), we feel quite confident that we did not miss a population of cells that activated TCF/LEF signaling, but did not respond to the TCF-Trace labeling system we devised.

15) R#3 11. *In Figure 2G-J, almost all 16TF-VNP+ cells are PAX7+ at E6.5; therefore, how do the Authors explain the high proportion of PAX7+ cells being TCF-Trace negative at E16.5, as shown in Figure 4K-L?*

Response: It is correct that at E6.5 almost all 16TF-VNP+ are PAX7+. However, this is a dynamic population that actively contributes to the formation of the primary myofibers. At E7.5, for instance, most, if not all myofibers have integrated nuclei derived from the 16TF-VNP+ population (Figure 4 B). By E16.5, the PAX7+/16TF-VNP+ progenitor population is minor. Our current model is therefore that this progenitor population is exhausted through its massive participation to the formation of the primary myofibers population and that the vast majority of PAX7+ muscle progenitors present at this stage derive from the 16TF-VNP-lineage.

16) R#3 12. *In Figure 5J, why is there no dTomato expression as compared to 5K? The data provided on CXCR4 is preliminary and is not sufficient to conclude that it is the only pathway regulating the spatial distribution of primary progenitors.*

Response: To evaluate the expression of *Cxcr4*, we electroporated brachial somites with the CAGGS:dTomatoNLS alongside with the CAGGS:DN LEF1 plasmid. As electroporation results, at best, in 50% of the total cells electroporated, dTomato+, electroporated myoblasts co-exist with non-electroporated, dTomato- myoblasts. Fig5J is a representative image of non-electroporated cells, immunostained for PAX7 and hybridized with the *Cxcr4* probe. As PAX7 is a transcription factor, it is expressed in the nucleus of muscle progenitors, and we therefore used this staining to delineate the nucleus of non-electroporated muscle progenitors. Fig5K is a representative image of electroporated cells with a plasmid coding for a nuclear dTomato alongside the dominant negative form of LEF1, immunostained for PAX7 and hybridized with the *Cxcr4* probe. The same strategy has been used in non-electroporated cells. We therefore compared the number of *Cxcr4* dots, in non-electroporated and electroporated cells. We added these precisions to the figure legend and the methods (lines 682-701, 899-900).

Addressing a comment of reviewer #1, we now provide further evidence, derived from the human limb embryo data published recently, that *CXCR4* is specifically expressed in the primary and not the secondary lineage (new Supplementary Figure 13), and described in the text (lines 420-435), reinforcing the hypothesis that it is a *bona fide* transcriptional target of TCF/LEF signaling, as demonstrated in Figure 5K.

However, we totally agree with this reviewer that the TCF-CXCR4 connection we discovered is likely not the only pathway regulating the migration and/or distribution of primary progenitors. A number of classical studies have demonstrated that muscle progenitor migration in the limb is a complex process, relying on the function of a number of genes acting at different levels of the process. Among those, PAX3 (Goulding et al., Development 1994), the receptor c-Met and its ligand SF (Bladt et al. Nature 1995), Lbx1 (Schäfer and Braun, Nature Genet. 1999) and CXCR4 (Vasyutina et al. Genes & Dev. 2005). The phenotype obtained after loss of function of CXCR4 is in fact the least impressive of the group, with a muscle distribution phenotype that is only temporary. It is possible that the function of receptor/ligand couple CXCR4/SDF-1 is redundant with or superseded by that of c-Met/SF. We hope this reviewer will agree with us that finding the exact function of TCF-*Cxcr4/Sdf-1* mini gene-regulatory network we uncovered during muscle progenitor migration and distribution is beyond the scope of this study.

17) R#3 13. *The replicate numbers and statistical methods used should be clearly described.*

Response: We added these details in the Methods section (lines 708-719)